



# Trace gas composition in the Asian summer monsoon anticyclone: A case study based on aircraft observations and model simulations

5    Klaus-D. Gottschaldt[1], Hans Schlager[1], Robert Baumann[1], Heiko Bozem[2], Veronika Eyring[1], Peter   Hoor[2], Patrick Jöckel[1], Tina Jurkat[1], Christiane Voigt[1,2], Andreas Zahn[3], Helmut Ziereis[1]

[1]Deutsches Zentrum für Luft- und Raumfahrt (DLR), Institut für Physik der Atmosphäre, Oberpfaffenhofen, Germany

10    [2]Johannes Gutenberg-Universität, Institut für Physik der Atmosphäre, Mainz, Germany

[3]Karlsruher Institut für Technologie (KIT), Institut für Meteorologie und Klimaforschung, Karlsruhe, Germany

*Correspondence to*: Klaus-D. Gottschaldt (klaus-dirk.gottschaldt@dlr.de)



**Abstract.** We present in-situ measurements of the trace gas composition of the upper tropospheric (UT) Asian summer monsoon anticyclone (ASMA) performed with the High Altitude and LOng range (HALO) research aircraft in the frame of the Earth System Model Validation (ESMVal) campaign. Air masses with enhanced $O_3$ mixing ratios were encountered after entering the ASMA at its southern edge at about 150 hPa on 18 September 2012. This is in contrast to previous studies, reporting that the anticyclone's interior is dominated by recently uplifted air with low $O_3$ in the monsoon season. We also observed enhanced CO and HCl in the ASMA, tracers for boundary layer pollution and tropopause layer (TL) air or stratospheric inmixing, respectively. In addition, reactive nitrogen was enhanced in the ASMA. Along the HALO flight track across the ASMA boundary, strong gradients of these tracers separate anticyclonic from outside air.

Lagrangian trajectory calculations using HYSPLIT show that HALO sampled three times a filament of UT air, which included air masses uplifted from the lower or mid troposphere north of the Bay of Bengal. The trace gas gradients between UT and uplifted air masses were preserved during transport within a belt of streamlines fringing the central part of the anticyclone (fringe), but are smaller than the gradients across the ASMA boundary. Our data represent the first in-situ observations across the southern and downstream the eastern ASMA flank, respectively. Back-trajectories starting at the flight track furthermore indicate that HALO transected the ASMA where it was just splitting into a Tibetan and an Iranian part. The $O_3$-rich filament is diverted from the fringe towards the interior of the original anticyclone, and at least partially bound to become part of the new Iranian eddy.

A simulation with the ECHAM/MESSy Atmospheric Chemistry (EMAC) model is found to reproduce the observations reasonably well. It shows that $O_3$-rich air is entrained by the outer streamlines of the anticyclone at its eastern flank. Back-trajectories and increased HCl mixing ratios indicate that the entrained air originates in the stratospherically influenced TL. Photochemical ageing of air masses in the ASMA additionally increases $O_3$ in originally $O_3$-poor, but CO-rich air.

Simulated monthly mean trace gas distributions show decreased $O_3$ in the ASMA centre not in general, but only at the 100 hPa level in July and August, as also reported by previous studies. However, at lower altitudes and in September the ASMA is dominated by increased $O_3$, indicating that the above processes are more important for the ASMA trace gas budgets than previously thought.



## 1 Introduction

The Earth System Model Validation (ESMVal) field experiment with the High Altitude and Long Range Research Aircraft (HALO, http://www.halo.dlr.de) was conducted during 10 - 24 September 2012 in close cooperation with the HALO TACTS mission (Jurkat et al., 2014; Vogel et al., 2015; Müller et al., 2015). During the 65 HALO flight hours of the ESMVal campaign, trace gas distributions were sampled from the ground to a maximum altitude of 15.3 km along the following route: Oberpfaffenhofen (Germany) – Sal (Cape Verde) – Cape Town (South Africa) – boundary of Antarctica – Cape Town – Male (Maledives) – Larnaca (Cyprus) – Oberpfaffenhofen – Spitzbergen (Norway) - Oberpfaffenhofen. The goal was to gather in-situ observations for the evaluation of Earth system models and to improve process understanding. Specific areas of interest included regions impacted by deep convection, lightning and biomass burning in West- and South Africa, anthropogenic pollution in Europe and the Mediterranean, the northern and southern polar regions, and the North African and Asian monsoons.

The Asian summer monsoon (also known as Indian or south west monsoon) sensu stricto is a prevailing sea breeze, lasting from June to September. Different mechanisms may contribute to the formation of a conduit of rising air, centered over the southern Tibetan plateau (Bergman et al., 2013) during northern summer. A high pressure area forms in the convective detrainment altitudes, sustaining a coherent anticyclone, centered at 200 to 100 hPa (Dunkerton, 1995; Randel and Park, 2006; Garny and Randel, 2016).

Polluted boundary layer air is entrained from throughout the region. It is effectively uplifted first in the narrow conduit to detrainment altitudes of about 200 hPa, later by large scale upward motion at the eastern side of the anticyclone and then confined by the Asian summer monsoon anticyclone (ASMA) (Lelieveld et al., 2001; Li et al., 2005; Randel and Park, 2006; Park et al., 2007; Park et al., 2008; Park et al., 2009; Chen et al., 2012; Bergman et al., 2013; Vogel et al., 2015; Ploeger et al., 2015). As a consequence, trace gas mixing ratios within the anticyclone are mainly shifted towards lower tropospheric values, e.g. relatively increased carbon monoxide (CO) (Li et al., 2005; Park et al., 2008) and decreased ozone ($O_3$) (Randel and Park, 2006; Park et al., 2007; Park et al., 2008; Kunze et al., 2010). The in-situ measurements considered in our study also show enhanced CO mixing ratios in the ASMA, but no decreased $O_3$. On the contrary, all ESMVal measurements in the ASMA show significantly increased $O_3$ mixing ratios - relative to the UT south of the anticyclone.

While several studies looked into the boundary layer sources for ASMA air, entrainment of stratospheric or TL air has received much less attention. We are not aware of a study focusing on that, although the possibility of stratospheric entrainment at the eastern flank of the ASMA has already been recognized (Plumb, 2005; Randel and Park, 2006; Ren et al., 2014). Other studies (Cristofanelli et al., 2010; Barret et al., 2016) indicate the almost absence of stratospheric intrusions during the monsoon season. At least the entrainment of $O_3$-rich TL air is supported by the HALO ESMVal in-situ observations considered here. Enhanced $O_3$ was also found in CARIBIC (http://www.caribic-atmospheric.com) in-situ measurements in the monsoon region. Trace gas signatures from the northern part of the ASMA were interpreted as photochemically older than those from the more central region, sampled at the southernmost parts of the CARIBIC flights (Baker et al., 2011; Rauthe-





Schöch et al., 2016). However, the origin of enhanced $O_3$ in the old air is not entirely clear, and Baker et al. (2011) also mentioned the possibility of stratospheric influences. Whether $O_3$ in air originating in the ASMA is generally enhanced or depleted was pointed out as one of the major open questions related to the Asian monsoon already by Lawrence and Lelieveld (2010).

Here we present a unique set of upper troposphere (UT) in-situ measurements in the ASMA, obtained during the HALO ESMVal campaign. The focus is on the measurements, their representation in a global chemistry climate simulation and the origin of air masses. We pinpoint the processes that led to the observed trace gas signatures by a more detailed analysis of an exemplary flight segment, and conclude showing how the present measurements could be reconciled with seemingly contradictory previous studies. In a follow-up study the

processes that determine the ASMA composition are further analyzed by putting the HALO ESMVal measurements into a regional, seasonal and multi-annual perspective. We refer to that study (Gottschaldt et al., 2016) as accompanying paper in the following. The accompanying paper is mainly based on EMAC simulations, which also show that our in-situ data reflect rather common processes in the ASMA. Both studies shall help to explain the highly variable composition of the ASMA and its outflow, addressing the following key aspects of

the ASMA that were recently identified as poorly understood (Randel et al., 2016): Dynamical and chemical coupling with convection, composition/reactive chemistry in the monsoon region, mixing of higher latitude lower stratospheric air into the tropical TL by the ASMA.

The paper is structured as follows: Section 2 provides a description of the instruments and techniques used for the in-situ measurements of selected tracers during the ESMVal campaign, the Eulerian global chemistry climate

simulation hindcasting the synoptic situation of the measurements, and the trajectory model used. The transport pathways of air masses that contributed to the observed chemical composition and periods of interest from the measured time series are identified in section 3. Section 4 shows that the EMAC global simulation may be used for the interpretation of the in-situ measurements, because the main features are reproduced well. Section 5 is dedicated to the discussion of selected tracer-tracer relations in the in-situ data. The eastern flank of the ASMA

is found to be crucial for the generation of the observed trace gas signatures, which is discussed in section 6 for that part of the flight providing the most direct observations of it. In section 7 we reconcile the HALO ESMVal observations of increased $O_3$ with previous studies that found decreased $O_3$ in the ASMA, and then conclude with a summary in section 8.

## 2 Methods

### 2.1 In-situ measurement techniques

All in-situ data used in our analyses are based on a synchronized data set, created by merging the data with their individual time resolution to a common time axis with a resolution of 10 seconds. This data set is available from the HALO database (https://halo-db.pa.op.dlr.de).





Carbon monoxide (CO) was measured with the three channel quantum cascade laser infrared absorption spectrometer TRIHOP. CO data were sampled every eight seconds with an integration time of 1.5 seconds and a total 1-sigma-uncertainty of 1.8 nmol/mol (Schiller et al., 2008; Müller et al., 2015).

A newly designed atmospheric chemical ionization mass spectrometer (AIMS) with an electrical discharge source and in-flight calibration provided HCl mixing ratios with a 1σ detection limit of 10–15 pmol/mol and an accuracy of 12 % (Jurkat et al., 2014; Voigt et al., 2014; Jurkat et al., 2016). AIMS measurements compared in general well to remote sensing techniques on board of HALO, like the limb sounding Michelson interferometer GLORIA (Ungermann et al., 2015).

Total reactive nitrogen, $NO_y$, is the sum of all reactive nitrogen species in the atmosphere. Besides $NO_x$ (= NO +
$NO_2$), $HNO_3$, peroxyacetyl nitrate (PAN), HONO, $N_2O_5$, $HO_2NO_2$, $NO_3$ are the most abundant species of the odd nitrogen family. NO and $NO_y$ were measured during the ESMVal campaign by a two channel NO-chemiluminescence detector (ANEAS) in combination with a gold converter installed in one channel (Ziereis et al., 2000). The detection limit is about 8 pmol/mol for an integration time of 1 s. Its overall uncertainty is about 8 % (6.5 %) for volume mixing ratios of 0.5 nmol/mol (1 nmol/mol).

The $O_3$ instrument FAIRO deployed during the ESMVal campaign is based on a chemiluminescence sensor plus an UV photometer  (Zahn et al., 2012), achieving at least 1.5% precision.

## 2.2    Atmospheric dynamics and chemistry simulations

The Eulerian simulation used for this study was performed with the EMAC model (Jöckel et al., 2010) within the project "Earth System Chemistry integrated Modelling" (ESCiMo), as a German contribution to the Chemistry
Climate Model Initiative (CCMI). This simulation has been described and evaluated in detail by Jöckel et al. (2016) as RC1SD-base-10a. Its setup is based on the CCMI transient hindcast reference simulation in specified dynamics mode (Eyring et al., 2013). Here we refer to it just as "simulation". Key characteristics of the simulation are a representation of the global domain with a spherical truncation of T42 and 90 vertical hybrid pressure levels up to 0.01 hPa, nudging of dynamics towards ERA-Interim re-analyses data (Dee et al., 2011)
from the free troposphere to a pressure altitude of 10 hPa. The simulation also includes complex interactive chemistry with on-line feedback on dynamics. This setup reproduces the synoptic situation during the aircraft campaign, allowing direct comparisons of simulated and measured data. In particular it was shown by Nützel et al. (2016) that the climatological representation of key dynamical features of the ASMA in ERA-Interim is in line with most other re-analysis data sets. However, some processes can not be explicitly resolved on the coarse
grid of the global simulation. These sub-grid scale processes are accounted for by parameterizations that are expected to reproduce climatological characteristics rather than individual events. This pertains in particular to convection and convective transport, thereby negatively affecting the non-climatological comparability of simulated versus observed atmospheric trace gas mixing ratios (Tost et al., 2010). Global fields are output with a frequency of 10 h, but data along the HALO flight track were sampled with the model time step resolution of 12
35  min (see description of S4D by Jöckel et al. (2010)). In addition to the nitrogen oxides listed in section 2.1, the EMAC tracers NHOH, HNO, $NH_2O$ and $NH_2OH$ are also included in simulated $NO_y$. It is calculated as the





mixing ratio of nitrogen atoms in the sum of the listed $NO_y$ compounds, which is consistent to what is measured by the corresponding in-situ instrument. Net photochemical $O_3$ production rates are calculated from the diagnostic tracers ProdO3 and LossO3 (Jöckel et al., 2016). Unless stated otherwise, the tropopause is diagnosed in our EMAC simulations according to the WMO definition between 30°S and 30°N, and by PV = 3.5 PVU

elsewhere.

### 2.3    Lagrangian trajectory calculations

The HYbrid Single-Particle Lagrangian Integrated Trajectory (HYSPLIT) model (Draxler and Hess, 1998; Draxler and Rolph, 2015) was used to calculate backward-trajectories, starting at the flight track. The same starting time is used for all back-trajectories of the selected flight segment, i.e. the time lag between different

positions along the track is neglected. Thus smaller flight segments need to be chosen, if the wind fields are more dynamic. HYSPLIT is driven kinematically, by meteorological fields of the Global Data Assimilation System (GDAS), in 1° x 1° horizontal resolution, 23 vertical levels between 1000 and 20 hPa, as provided by NCEP (National Weather Service's National Centers for Environmental Prediction, http://ready.arl.noaa.gov/gdas1.php) at 3-hour time steps. Convection is represented by this approach indirectly, as smoothed vertical velocity

components in the reanalysis fields. HYSPLIT trajectories only capture advection and stirring. Note that in contrast, EMAC captures convection directly (although parameterized), and also mixing in the form of diffusion (Roeckner et al., 2006).

### 3    Air masses observed and transport pathways

An air mass with enhanced mixing ratios of $O_3$, CO, NO, $NO_y$ and HCl (Fig. 1) was sampled during a flight on 18 September 2012 from the Maldives to Cyprus over the Arabic Sea. Here, HALO flew at an altitude of 160 to 170 hPa, just before reaching the Oman coast (Fig. 2). We attribute the sudden increase of the above mentioned trace gases to the entering of the ASMA from the south. The sampling of this air mass was interrupted by a dive to probe the lower boundary of the ASMA, but after that HALO continued to fly in UT air related to the ASMA.

In this study we focus on the interpretation of the measurements in those clearly ASMA-related air masses, marked yellowish in the left column of Fig. 1, and indicated by three black ovals in Fig. 2a. We divide the ESMVal-flight from Male to Larnaca pragmatically into 7 parts (Table 1), called periods of interest (POI) in the following. We refer to the central region as "interior", and to the boundary region, i.e. the outer streamlines of the ASMA circulation, as "fringe". The terms "interior" and "fringe" characterize actual positions of streamlines

within the anticyclone, independent of the trace gas signatures they carry.

POI1: The first part of the flight consists of the take-off from Male and ascent. It is not directly related to the ASMA and not further discussed here.

POI2: HALO was flying in the UT in a north-westerly direction towards the Arabic peninsula. POI2 ends south-east of the Oman coast with the sudden increase of measured $O_3$ and other trace gas mixing ratios (Fig. 1). The

corresponding back-trajectories (supplementary material) indicate that air masses came from the Far East Pacific





coast, from the boundary layer of South East Asia, and from the ASMA. Mixing ratios of $O_3$, CO, HCl and nitrogen oxides are significantly decreased compared to the following flight segment, indicating that POI2 is dominated by clean air. We do not consider this adjoining air as part of the ASMA, although it may get entrained occasionally. The back-trajectories for this flight segment depend critically on the HYSPLIT start time, indicating very dynamic wind fields and possibly inaccurate trajectories. This might also be the reason that EMAC simulation results for some tracers showed different gradients than observed. We therefore decided not to analyse this flight segment in greater detail here, despite it might be interesting for characterizing the southern boundary region of the ASMA.

POI3: The next flight segment (indicated by "3" in Fig. 2a) is characterized by almost parallel back-trajectories along the southern ASMA fringe (Fig. 3c). The outer trajectories show air masses circling around the ASMA within 10 days (Fig. 3a) at 160-170 hPa (Fig. 3b), while the inner trajectories were first uplifted at the southern/south-western flanks of the Himalayas, then the Tibetan conduit to merge with the UT ASMA circulation at its eastern flank (Fig. 3d). The back-trajectories of POI3 mainly encompass South Asia and the Arabic peninsula.

POI4: Back-trajectories from the following dive over Oman are given in the supplementary material. The dive was intended to explore the vertical structure of the ASMA and its lower boundary. However, almost immediately below the flight altitude of POI3 the back-trajectories no longer clearly indicate direct transport of air from the eastern ASMA flank. The flight segments at the beginning and at the end of the dive, each covering the altitude range from about 180 hPa to 400 hPa, show a more or less curled-in structure similar to POI5 (Fig. 4) near the HALO track. This indicates complex stirring. The anticyclonic motion in the upper parts of POI4 is much slower than in the ASMA above, which was sampled during neighbouring POI3/5: Air masses of POI5 travelled almost twice as far as those of POI4 within 11 days (Fig. 3e vs Fig. S4e). Just for reproducibility: The lower boundary of the ASMA was defined as the region where no back-trajectory was circling the entire ASMA and reaching the southern flank within 280 h anymore. There is almost no transition between fast and slow air masses, indicating a sharp lower boundary of the ASMA in terms of wind speeds. However, our pragmatic criterion might not work in general and should not be applied to other cases without further analysis. The two high-altitude parts of POI4 might in principle be useful for characterising the lower boundary of the ASMA, but that would require a dedicated analysis and is not the focus of our study. At lower altitudes (~400 hPa to 650 hPa) the origin of air encountered during the dive shifts towards the Mediterranean and Europe, which is also not further analysed here.

POI5 and POI6 (POI5/6): The next two flight segments lead from the Persian Gulf to the Eastern Mediterranean ("5" and "6" in Fig. 2a). Like POI3, they are characterized by almost parallel streamlines from the eastern ASMA flank, along the southern fringe, to the measurement location. The filament of UT air was curled in horizontally in such a way that the (former) ASMA fringe was transected during POI5 (Figs. 3efgh) outside-in, then inside-out during POI6 (Figs. 3ijkm). This complex structure of the streamlines is a consequence of an ASMA splitting or eddy shedding event that occurred during the ESMVal campaign (discussed in more detail in the accompanying study). The original ASMA encompassed South Asia and the Arabic peninsula (Figs. 3abc).





HALO crossed the zone where the original ASMA separated into two smaller anticyclones, one centred over the Iranian plateau and one centred over Tibet. The air masses contributing to POI3/5/6 all passed the eastern ASMA edge over South Asia, but at least POI5 is bound to become part of the Iranian anticyclone after the splitting. The transition from POI5 to POI6 was chosen according to the initial direction of back-trajectories changing from right to left of the HALO track. We regard the transition from POI5 to POI6 as a pragmatic estimate only, as it is essentially the same air mass. According to a zoom (not shown) the back-trajectories were also curled in vertically at the transition from POI5 to POI6 to some degree, indicating complex small scale dynamics. The back-trajectories for POI5/6 show that the anticyclonic motion encompassed northern Africa earlier in September 2012, which also applies to POI2. However, unlike for POI2 it did not extend far into Eastern Asia. UT air contributing to POI3/5/6 passed the eastern ASMA flank mainly over South Asia.

POI7: The descent into Larnaca shows some similarities to the dive over Oman. As in POI4, the anticyclonic motion becomes slower with decreasing altitude and the air encountered above about 400 hPa has a mainly south Asian origin. Back-trajectories starting at the flight track at the north-western flank of the ASMA below a pressure altitude of about 130 hPa do not encircle the anticyclone within 280 h. In contrast, those starting at higher altitudes do. POI7 is not further discussed here, but might be considered in future studies to characterise the lower ASMA boundary.

During POI3 the fringe was transected outside-in. Deep convection at the eastern ASMA flank contributed considerably to POI5 (Fig. 3h), and very little to POI6 (Fig. 3m). POI3 passed the eastern ASMA flank on 15 September (Fig. 3a), POI5 on 12 September (Fig. 3e), POI6 on 13 September 2012 (Fig. 3i). Note that the eastern ASMA flank moved eastward during that period, and the area enclosed by the back-trajectories did shrink (Figs. 3aei). A schematic of the synoptic situation for POI3/5/6 is given in Fig. 4. All three POIs are part of a filament that spent at least 10 days in the UT of the ASMA region, and was entrained by updrafts at the eastern ASMA flank. The curled-in structure of the filament indicates that the ASMA split into a Tibetan and an Iranian part around 18 September 2012.

Six-hourly satellite images show no signs of fresh convection in the vicinity of the HALO track on the days before the flight (see supplementary material). Shorter-lived, localized convective events were identified in 15 minute satellite images (not shown) over the Hajar mountains (Oman) and east of the Strait of Hormuz on 17 and 18 September 2012, afternoon. We set a more detailed discussion of this aspect aside here, since dispersion calculations (not shown) indicate that the UT ASMA measurements during the ESMVal campaign (POI3/5/6) were not affected by those convective plumes. Furthermore, videos from the cockpit camera show that HALO did not transect the convective region over the Hajar mountains on 18 September 2012. Highly polluted air (e.g. increased $SO_2$) from the Persian Gulf (McLinden et al., 2016), uplifted the day before, may have affected the dive (POI4) though.

POI3 is less affected by stirring during transport from the eastern ASMA flank to the measurement location than POI5/6 and thus provides a more direct view of the remote and so far unsampled eastern flank. In addition to stirring, diffusion may also act to conceal features of trace gas distributions during transport. However, assuming a diffusion coefficient of 15 $m^2s^{-1}$ (Schumann et al., 1995), purely diffusive mixing is negligible here. It acts on a





scale of about 1 km per day, and air parcels needed less than a week from the eastern flank to their respective measurement locations. For comparison, two measurement points in 10 s time resolution at typical HALO speed are about 2.5 km apart.

We also note that the outer ASMA edge was only crossed during the beginning of POI3, and we pragmatically chose a steep gradient of $O_3$ mixing ratios to distinguish the ASMA from outside air. The flight path could have transected the former outer ASMA edge two more times, due to the curled-in structure of the filament (Fig. 4). However, HALO dived below the ASMA at the beginning of POI5 and at the end of POI6. Measurements at the original flight level might have provided a more natural definition of the ASMA boundary, because the separation of ASMA air from outside air had already taken place for those older parts of the filament. In contrast, it is not clear if the steep $O_3$ gradient, chosen as beginning of POI3, corresponds to where outside flow later separated from the ASMA circulation. Thus we do not attempt to estimate, if/how much outside air becomes part of the ASMA circulation by entrainment at the southern edge. This uncertainty is not important for the present study, but might need to be addressed before quantitatively estimating trace gas budgets within the ASMA.

## 4    Representation of the in-situ measurements in EMAC

Here we discuss to what extent the aforementioned (section 2.2) simulation with the EMAC model can reproduce the $O_3$, HCl, CO, NO and $NO_y$ measurements in order to use that simulation for further interpretation of the measurements.

The EMAC simulation has a horizontal grid resolution of about 300 km in the ASMA region, and the time step length is 12 min. Processes acting on smaller, unresolved scales need to be parameterized. This is compared to in-situ data with a time resolution of 10 s, corresponding to a spacing of about 2.5 km. Due to the different resolutions, a perfect match between simulation and observations can not be expected. Additional differences might be caused by non-perfect representations of emissions, physical and chemical processes.

EMAC in the setup used here is known to simulate a high $O_3$ bias in the tropics, more specifically of 5-25% at 100 - 250 hPa, compared to ozone sonde data, and 30-50% in the tropospheric column compared to satellite data (Jöckel et al., 2016). However, the relative enhancement observed during the POIs is reproduced by the simulation (Fig. 1a).

The HCl mixing ratios encountered during the flight from Male to Larnaca were at the detection limit of the AIMS instrument, therefore with enhanced noise. They were interrupted by missing value periods due to calibrations and background measurements. In order to carve out variations on a time scale relevant for this study, we smoothed the HCl in-situ data as follows: Each original value of the time series is substituted by the average of the mean of all values 150 s before, and of the mean of all values 150 s after the original value. Missing values are ignored and each operation is based on equal weights. This procedure gives values in periods of sparse data greater weight, but was found to preserve the shape of the time series better than a conventional running mean filter. Note that the time series technically is still in 10 s resolution after the smoothing, but with regard to contents, time resolution has been traded for a better signal-to-noise ratio. In-situ measurements in the





subtropical UTLS over North America (Marcy et al., 2004) indicate that UT background mixing ratios of HCl may be in the order of 5 pmol/mol. Such low values are found in our data (Fig. 1c) during clean-air-dominated POI2 and in the middle of POI5, where back-trajectories point to lower tropospheric air (Figs. 3gh). Relative HCl enhancements in other sections of the flight are also clearly visible in Fig. 1c, indicating inmixing of stratospheric or TL air then. The simulation reproduces the magnitude of measured HCl, and roughly also the time evolution during the POIs. We consider the agreement as reasonable, given the uncertainties of the measurements, as well as the possibility of spurious washing out and slightly misrepresented gradients of trace gas mixing ratios (Fig. 1d, Fig. 2b) in the simulation. The relative minimum in free tropospheric HCl is best seen in the curtain (Fig. 1d), together with a filament of increased HCl extending from the tropopause to the flight track around 8 UTC.

Considering the use of monthly instead of daily resolved biomass burning emission data, there is a surprisingly good agreement between measured and simulated CO. The air masses encountered during the measurements might have experienced sufficient mixing since last boundary layer contact to lose memory of any high frequency emission variations, making monthly emissions in the simulation a viable approximation here. There is a negative bias of simulated CO of about 10 nmol/mol during POI5/6 (Fig. 1e). Figure 2d shows that POI5/6 coincide with a region of strong CO gradients. This may result in some inaccuracies in the simulated values along the flight path, even if the synoptic situation is captured well by the simulation. Uncertainties in the chemical mechanism also have the potential to cause a low bias of CO and a high bias of $O_3$ (Gottschaldt et al., 2013; Righi et al., 2015). In any case, caution is needed when interpreting those measurements based on the simulation. We focus on the best represented flight section (POI3) whenever possible.

The relative changes of observed $NO_y$ (Figs. 1ij) and NO (Figs. 1gh) mixing ratios are captured by the simulation, in particular the enhancements of those trace gases in the ASMA. However, observed short time scale variations during POI5/6 are smoothed out in the simulated data by the coarser output (time) resolution directly, and indirectly because the representation of processes is limited by the grid resolution. The representation of nitrogen oxides in the simulation also depends on the quality of the corresponding emission inventories, and is further complicated by the shorter photochemical lifetime compared to CO and $O_3$. NO – and to a lesser degree also $NO_y$ – mixing ratios have steep vertical gradients at the flight altitude during POI5/6 (Figs. 1hj). We do not expect a global simulation to perfectly reproduce time and location of such features, and the corresponding inaccuracies are most likely to print through in the vicinity of steep gradients. Also, parameterizations of sub-grid scale processes are mainly designed to reproduce climatological characteristics and individual convective events in the simulation may not be triggered at the same times and locations as in reality. There are regions of over- and underestimated NO and $NO_y$, respectively, and we don't expect any systematic bias in the representation of nitrogen oxides in the simulation. In particular, we note that the magnitude of measured NO mixing ratios is reproduced by the simulation, although most UT $NO_x$ in the ASMA is produced by lightning (see accompanying paper for details) and estimates of lightning $NO_x$ emissions include large uncertainties (Schumann and Huntrieser, 2007).





Summarizing, the limited resolution of the simulation is at the core of most of the deviations between observed and simulated trace gas mixing ratios. This means in return that a synoptic scale feature like the ASMA is likely to be represented well by the specified dynamics simulation setup, which is carried forward to the related large scale trace gas distributions. Furthermore, intensity, frequency and localization of convective activity in the region (Bergman et al., 2013) might not leave too much freedom to EMAC's convection parameterization to deviate from reality in this particular case. Overall, we are confident that the simulation reproduces the atmospheric situation well enough to be utilized for interpreting the in-situ data of the POIs. The overall agreement between observed and simulated data is best for POI3.

## 5    Tracer-tracer relations

Enhanced tropospheric tracers (CO) fit the climatological picture of the ASMA, but at the same time enhanced $O_3$ and HCl is notable and indicate enhanced in-situ production, or contributions of stratospherically affected air e.g. from the TL. In this section we determine the origins of the measured trace gas signatures with the help of tracer relations. The following analysis focuses on CO vs $O_3$, and is supplemented by other relations (HCl vs $O_3$, $NO_{(x)}$ vs $O_3$, $NO_x$ vs $NO_y$). POI3, POI5 and POI6 are part of one filament, and all are characterized by mixing of UT air with uplifted lower tropospheric air at the eastern ASMA flank (Fig. 4). In the following, we exemplify tracer-tracer relations in the filament by a discussion of POI3. We focus on that period, because it is best represented in the simulation and the dynamics is less complicated than for POI5 and POI6. The latter means that POI3 provides the most direct view of the eastern flank and the relevant processes of that key region are least concealed by stirring. Furthermore, the air encountered during POI5/6 has more remote source regions (Figs. 3ei) and was subject to longer transport since passing the eastern ASMA flank. Thus, it is easier to disentangle the relevant processes for POI3.

### 5.1    Mixing of different reservoirs during POI3

All measurements from 7:47 to 8:15 UTC (dots) lay on one mixing line in the CO vs $O_3$ plot for POI3 (Fig. 5a). It connects a CO-poor & $O_3$-rich reservoir (CO↓↑$O_3$) with a CO-rich & $O_3$-poor reservoir (CO↑↓$O_3$). The "&" notation is used in the following to express "and at the same time". Rich and poor are meant relative to the ranges observed during that flight section. Park et al. (2007) proposed thresholds of CO > 60 nmol/mol and $O_3$ < 300 nmol/mol to characterize tropospheric air in the ASMA. According to that criterion, absolute mixing ratios of POI3 are completely tropospheric. However, the lowest $O_3$ mixing ratios in POI3 (about 68 nmol/mol) still represents a significant enhancement with respect to 30 nmol/mol at the end of POI2 and also compared to 40 nmol/mol encountered below the ASMA during POI4 (Fig. 1a). The term reservoir is used here for the current state rather than for hypothetical end members. The negative slope indicates either an ozone-depleting photochemical regime (Baker et al., 2011), or that both reservoirs have seen different stratospheric or TL influences. The latter is supported by relatively enhanced HCl mixing ratios in CO↓↑$O_3$ and accordingly depleted HCl at the end of CO↑↓$O_3$ (Fig. 5c). A positive correlation between $O_3$ and HCl inside the ASMA has also been found by (Park et al., 2008), based on ACE-FTS satellite data, and attributed to a common





stratospheric origin of both species. The negative correlation between CO and $O_3$ in the ESMVal in-situ data is consistent with MLS observations in the ASMA region (at 215 hPa) that cover multiple entire years (Livesey et al., 2013).

According to Figs. 3abcd, trajectories carrying signature $CO{\downarrow}{\uparrow}O_3$ make up the outer fringe of the ASMA, travelling along almost closed streamlines at an altitude of about 160 hPa, which is in the tropopause region on the northern flank, and well in the troposphere on the southern flank of the ASMA. The inner (with respect to ASMA) streamlines of the filament are dominated by signature $CO{\uparrow}{\downarrow}O_3$. The corresponding back-trajectories indicate air masses uplifted from the boundary layer at the southern flanks of the Himalayas, and mid tropospheric air uplifted over the southwestern flanks of the Himalayas. Both meet at about 300 hPa over the Tibetan plateau, to be further uplifted to the UT, and merged with the anticyclone at its eastern flank. The line connecting $CO{\downarrow}{\uparrow}O_3$ and $CO{\uparrow}{\downarrow}O_3$ forms a linear correlation with a correlation coefficient of $r = 0.97$. This is very compact, indicating homogenous reservoirs. Freshly uplifted air did not seem to carry much small scale heterogeneity through the Tibetan conduit. This might explain that the simulation results compared rather well for this flight segment, because small scale differences between real and simulated (inventory) biomass burning could not print through to UT trace gas signatures. Independently of that, UT lightning $NO_x$ emissions still introduce differences between simulated and observed trace gas mixing ratios in the ASMA. Feeding of the inner trajectories of the filament through the Tibetan conduit defines reservoir $CO{\uparrow}{\downarrow}O_3$. Reservoir $CO{\downarrow}{\uparrow}O_3$ is air circling in the ASMA fringe in the UT. The mixing situation, as it occurred at the eastern ASMA flank, is carried by almost parallel streamlines to where it was encountered by HALO (Fig. 3c). Furthermore, shearing has been small in the air mass considered here, as indicated by little differential velocities (Fig. 3a). The strong correlation of the measurement time stamp (corresponding to the radial position in the fringe) with the location of the corresponding data in CO vs $O_3$ space is also indicative of almost parallel streamlines.

### 5.2 Processes reflected by nitrogen oxides during POI3

Consider the hypothetical case of a fixed $NO/NO_y$ partitioning (ratio): Variations of NO and $NO_y$ mixing ratios are reflected by positive slopes in NO vs $NO_y$ space then. Neighboring measurements indeed seem to lie on multiple parallel lines corresponding to NO proportional to $NO_y$ (indicated by grey lines connecting consecutive times / colors in Fig. 5b). Different lines correspond to different $NO/NO_y$ ratios though. The observed $NO/NO_y$ ratio decreases from about 0.33 in the outer streamlines of the filament (blueish dots throughout Fig. 5), to about 0.27 (reddish dots) at the end of POI1. As long as nitrogen species are not removed from the atmosphere, for example by rainout or washout, photochemical processes tend to convert NO to other $NO_y$ species and therefore change the $NO/NO_y$ ratio. Concurrently also $O_3$ and CO mixing ratios change along the transect of the filament (Fig. 5a). Increased $O_3$ (blueish dots, Figs. 5ad) is expected to shift $NO/NO_2$ photochemically towards $NO_2$, and $NO_2$ is part of $NO_y$. Thus increased $O_3$ should decrease the $NO/NO_y$ ratio by lowering NO. The opposite was observed: Increased $O_3$ corresponds to increased NO (Fig. 5d), while corresponding $NO_y$ mixing ratios are almost constant (Fig. 5b). The positive, linear correlation of NO vs $O_3$ measurements (Fig. 5d) might in itself be attributed to enhanced $O_3$ production due to increased NO. Such an interpretation would require a positive




correlation of NO vs $NO_y$ for the entire range of NO mixing ratios. However, positive correlations between NO and $NO_y$ were only observed for subsets of the data (grey lines in Fig. 5b). Summing up, we interpret the distribution of measurements in NO vs $NO_y$ space (Fig. 5b) as the overlay of small variations (noise, scatter) of nitrogen oxide mixing ratios on top of a decreasing $NO/NO_y$ ratio from outer streamlines towards more inside

the ASMA. The latter variation seems to be due mainly to mixing of reservoirs with different NO mixing ratios, namely a $NO{\downarrow}{\downarrow}O_3$ reservoir at inner streamlines with $NO{\uparrow}{\uparrow}O_3$ at outer streamlines. This is consistent with backward trajectories, which also indicate two different reservoirs (Figs. 3abcd). Lightning is the most likely source of increased NO in older UT ASMA air, as compared to NO-poor, freshly uplifted, air (see accompanying paper for details).

### 5.3    Synthesis for POI3 and related UT measurements

A decreased CO & decreased $O_3$ reservoir ($CO{\downarrow}{\downarrow}O_3$, indicated by crosses in Fig. 5) contributed to the outer ASMA streamlines, diluting increased $O_3$ signatures there. According to the backward trajectories, $CO{\downarrow}{\downarrow}O_3$ originates from mid tropospheric air, transported in cyclonic motion below the ASMA, then rapidly uplifted over Myanmar. A medium CO - increased $O_3$ reservoir ($CO{\uparrow}{\uparrow}O_3$) contributed to the inner edge of the filament,

which is mainly UT air originating from the interior of the ASMA. Note that both reservoirs with increased $O_3$ ($CO{\uparrow}{\uparrow}O_3$ and $CO{\downarrow}{\uparrow}O_3$) are not directly connected across parallel trajectories, since $CO{\uparrow}{\uparrow}O_3$ mixes towards the opposite end ($CO{\uparrow}{\downarrow}O_3$) of the central mixing line. The gradient between the inner and the outer edge might have been flattened by mixing during transport from the eastern ASMA flank to the measurement location, thereby just shortening the mixing line.

Figure 6 shows a schematic for POI3 and summarizes the main aspects discussed above. The fringe is essentially a transport barrier, separating the ASMA interior from the outside UT. Nevertheless, it interacts with the interior at its inner edge, and with the surroundings at its outer edge, resulting in trace gas gradients perpendicular to the streamlines of the fringe. The outer edge scrapes along the declining tropopause in the north and may entrain the TL when veering into the free troposphere at the eastern flank. Stratospheric intrusions from the tropopause

folding hotspot over the eastern Mediterranean (Akritidis et al., 2016) may potentially also contribute to the chemical composition of the fringe, but were not detected in the ESMVal measurements. Air uplifted from the lower and middle troposphere dominates the inner edge of the fringe. The trace gas signatures encountered by HALO before entering the fringe, and after leaving it towards the ASMA interior are again different, i.e. not the end members of the gradient in the fringe. Thus the fringe signatures must have been generated somewhere else,

most likely at the eastern flank.

The corresponding detailed tracer-tracer relations for POI5 and POI6 are shown in the supplementary material, but we do not discuss them in detail. Both belong to different sections of a filament of UT ASMA air that was more or less entrained by deep convection at the eastern ASMA flank. The mixing lines of POI6 may even be dominated by different amounts of inmixing from the tropopause region, rather than by air from the lower

troposphere. A detailed quantification of the different processes' contribution to individual measurements would however require more sophisticated analyses along back-trajectories.





## 6  Origins of observed ASMA trace gas signatures

### 6.1  Entraining the TL

During the POIs HALO was flying well below the tropopause. All observed tracer mixing ratios are clearly

tropospheric, and all back trajectory end points are in the troposphere. There is no indication of back-trajectories crossing the TP, which steeply slopes over the Tibetan plateau and is hard to define accurately there (Ren et al., 2014). Here it is only important to note that backward trajectories do not indicate any contribution from high above the tropopause region. This TL is subject to mixing, small scale stirring, convection (mainly tropical TL) and isentropic transport (mainly extra-tropical TL). All these processes involve cross-tropopause trajectories, but

our Lagrangian calculations would only capture large scale transport. The Eulerian EMAC simulations also reproduce large scale transport, but additionally capture small scale stirring/mixing as diffusive processes and convective transport, the latter controlled by the convection parameterization. There is a conspicuous filament of increased HCl & decreased CO signatures extruding from the tropopause trough at the eastern ASMA flank (black arrows in Figs. 2bd). This indicates entrainment of TL air, which has a more stratospheric signature as

compared to the surrounding upper tropospheric air. This is consistent with Randel and Park (2006), who inferred from a similar filament of increased $O_3$ (AIRS satellite data, 350 K isentropic level, 13 July 2003) that strong equatorward advection at the eastern flank of the ASMA might bring stratospheric air to low latitudes. The filament is hardly recognizable in the snapshot of $O_3$ from our simulation (Fig. 2a), because there are no big differences between $O_3$ mixing ratios in the fringe and those in the interior. Park et al. (2007) also found a

relatively high frequency of TL air at 100 hPa (MLS satellite data, July – August 2005) at the eastern ASMA flank, and interpreted this as an indicator for frequent stratosphere-troposphere exchange. Figure 3 shows that air in the fringe travels at almost constant altitude. It is scraping along the tropopause in the north, entraining the TL mixing zone. A filament with increased mixing ratios of HCl and $O_3$ is dragged into the troposphere (away from the TP) at the eastern ASMA flank, but at the same time the decreased $NO_y$ zone at the tropopause is not

disrupted (Fig. 2c). There is entrainment from the TL, but not from far above the TP. This process obviously contributes to increased HCl & $O_3$ mixing ratios in the ASMA fringe. A quantitative estimate of the amount of entrained $O_3$ like in Marcy et al. (2004) is however not feasible here, because all HCl measurements are at the instrument's detection limit. Moreover, the HCl-to-$O_3$ ratio in the source region of the entrainment would have to be known for such an analysis. Marcy et al. (2004) used stratospheric in-situ measurements from the same

region as their UT data, but the source region of our HCl is within the trace gas gradients of the South Asian TL. In-situ measurements across the TP at eastern ASMA flank would be desirable, but we are not aware of any such measurements from that region. In fact, POI3 of the HALO ESMVal campaign seems to provide the closest snapshot so far of trace gases from that interesting region. The trace gas signatures acquired by the outer ASMA streamlines at the eastern ASMA flank were carried almost unperturbed along the southern flank to the

measurement location. The preservation of trace gas gradients (Fig. 1, Fig. 5) indicates little mixing. In the following, we therefore take a closer look at the eastern flank as simulated by EMAC.




### 6.2    Air masses of POI3 at the eastern ASMA flank

The signature of the air mass observed during POI3 was foormed when it passed the eastern ASMA flank three days before the HALO flight (see 72 h back-trajectories in the supplementary material). The back-trajectories (Fig. 3b) also show that the UT part of air mass encountered during POI3 had been travelling at almost constant altitude for the time of circling the entire ASMA. Only when the flow is forced southwards at the eastern ASMA flank, trajectories briefly follow the steeply ascending TP (Fig. 3b: deep blue shadings in the NE part of the ASMA; Ascending trajectories in that region could alternatively be an artefact of convective activity, if expressed in the reanalysis data as smeared-out vertical velocity components). The trajectories, however, descend to their original altitudes after separation from the TP trough at the eastern ASMA flank, implying the existence of a flow component perpendicular to the TP when veering away. The flow field is also strongly divergent in the horizontal (see forward and backward trajectories from the eastern ASMA flank in the supplementary material). As a result, air from the TL is dragged southwards into the UT.

Figure 7 zooms into the simulated distribution of different trace gases on 15 September 2012, 8 UTC, at the eastern ASMA flank. A TP trough develops at the eastern flank at that time. TL air is entrained there into the ASMA fringe, as indicated by a filament with characteristic trace gas signatures: increased HCl (white arrow in Fig. 7a), decreased CO (Fig. 7d) and increased $O_3$ (Fig. 7g) – all relative to the surrounding UT air. Rising air from the Tibetan conduit arrives at the pictured altitude at the inner streamlines of the filament, contributing air enriched in CO, but depleted in HCl and $O_3$. This lower tropospheric air mass determines the inner streamlines of the ASMA fringe. Meridional curtains (middle column of Fig. 7) contain the center of the air mass that was encountered by HALO during POI3. Going back 72 hours from POI3, the HALO flight path corresponds approximately to an inclined zonal transect at 27°N (see supplementary material). Those simulated zonal transects (black bars in right column of Fig. 7) nicely reflect the trace gas gradients observed during POI3. We note that $O_3$ correlates with HCl, and both are anti-correlated with CO at the eastern ASMA flank. This is consistent with a common origin of increased-$O_3$ & increased-HCl signatures, supporting the hypothesis of TL contributions to the UT ASMA air. The plume of uplifted air that dominates the inner streamlines encountered during POI3, is in contrast characterized by increased CO and decreased HCl & $O_3$ (circled in Figs. 7beh).

The trace gas signatures of outer and inner streamlines of POI3 are shaped simultaneously at the eastern ASMA flank. However, the outer parts of the ASMA fringe that mainly entrained the TL at the eastern flank were only marginally part of the filament, which was transected by HALO. The air mass to be observed (centered at crosshairs in Fig. 7) was too far west of the eastern ASMA edge (arrows in Figs. 7agj). We cannot quantify from our analysis, how much TL entrainment contributed to the decreased-CO & increased-$O_3$ signature ($CO\downarrow\uparrow O_3$), measured at the beginning of POI3. Little scatter of the mixing line between $CO\downarrow\uparrow O_3$ and $CO\uparrow\downarrow O_3$ (Fig. 5a) indicates that only two reservoirs contributed, and that $CO\downarrow\uparrow O_3$ had been the signature of the fringe before arriving at the eastern flank already. In that case, the corresponding streamlines would not have acquired a modified signature at the eastern ASMA edge, but only have veered away from the TP there. Then the gradient between $CO\downarrow\uparrow O_3$ and $CO\uparrow\downarrow O_3$ during POI3 is a consequence just of adding $CO\uparrow\downarrow O_3$ to the inner edge of the filament. However, even if the TL air entrained by the more outer streamlines around 15 September (Fig. 7) did



not contribute to the HALO measurements, it still becomes part of the outer streamlines of the ASMA circulation. Increased HCl mixing ratios in the measurements indicate that earlier such events indeed have contributed to the trace gas signatures of the UT air that was already circling in the fringe before arriving at the eastern ASMA edge.

### 6.3   Photochemical $O_3$ production

How did photochemical $O_3$ production affect the $O_3$ gradient between inner and outer edge of the POI3-filament? According to EMAC, photochemical $O_3$ production is expected to be enhanced towards the ASMA interior (Figs. 7nop), where actually less $O_3$ was measured than at more outer streamlines during POI3. Increased $O_3$ mixing ratios were measured at the outer edge of the filament, where aged UT air dominates. Thus the measured mixing line cannot be explained by photochemical $O_3$ production after the air mass had been minted at the eastern ASMA flank. Contrary, integral photochemical $O_3$ production along the transport path acts to level the gradient.

The distributions of both main $O_3$ precursors, $NO_x$ and CO, print through as locally enhanced net $O_3$ production in Fig. 7n. Net $O_3$ production seems to depend more on CO (and related precursors) than on $NO_x$. Figures 7fmp again shows that $O_3$ production is maximal in the altitude range, where increased CO meets increased $NO_x$ mixing ratios. $NO_x$ is limited below, CO and volatile organic compounds above.

Could the high $O_3$ signature of the aged UT air (CO↓↑$O_3$) be due to in-situ photochemical $O_3$ production alone? Opposite gradients of net ozone production and ozone mixing ratios could also be explained by aged air (decreased $O_3$ production, increased $O_3$) circling in the fringe, and entrainment of young air (increased $O_3$ production, decreased $O_3$) at the inner edge of the filament. No TP entrainment would be needed in such a scenario of ageing uplifted air to explain increased $O_3$ mixing ratios. It is however inconsistent with the HCl and CO gradients in the transected filament, which have been observed (Fig. 5) and simulated (Fig. 2, 6). Thus entrainment from the TL – either on 15 September at the eastern ASMA flank or before - did contribute to the signature of the fringe filament encountered by HALO. Photochemical ageing certainly also has contributed to raise $O_3$, at least for nine days of circling in the fringe.

### 7   Is $O_3$ enhanced or decreased in the ASMA?

Based on the general picture provided by previous studies, we had expected to find decreased $O_3$ in the ASMA compared to the surrounding UT, but found increased mixing ratios instead. Thus either our presumption of generally decreased $O_3$ in the ASMA is wrong, or HALO encountered an unusual situation. In the following we revisit studies that advocated the picture of decreased $O_3$ in the ASMA (Randel and Park, 2006; Park et al., 2007; Park et al., 2008; Kunze et al., 2010; Kunze et al., 2016), while the frequency of occurrence of the processes needed to explain the in-situ data is further discussed in the accompanying paper.

Randel and Park (2006) and Kunze et al. (2010) base their analyses of $O_3$ on isentropic vertical coordinates, mainly the 360 K potential temperature level. Kunze et al. (2016) find a quite persistent pattern of decreased $O_3$





concentrations during strong monsoon seasons on the isentropic levels 360 K to 380 K. Isentropes tend to form a trough in the ASMA, when viewed in pressure coordinates, due to diabatic heating over the Tibetan Plateau (Ren et al., 2014). Thus it is more likely to find lower tropospheric trace gas signatures in the ASMA interior on potential temperature surfaces than on pressure levels. EMAC simulated $O_3$ in the ASMA is indeed decreased

on isentropic levels, but at the same time increased on various UT pressure levels in the same altitude range (see supplementary material).

However, Park et al. (2007) report an $O_3$ minimum in the ASMA at the pressure level of 100 hPa, for July and August 2005. That is based on MLS retrievals, which were recently found to have some low $O_3$ bias at around 100 hPa (Yan et al., 2016). Our simulation, nevertheless, reproduces an $O_3$ minimum at 100 hPa in the ASMA

for July and August, 2005 and 2012, respectively (see supplementary material). No $O_3$ minimum is simulated for September (2005, 2012), and neither for 150 hPa nor 200 hPa in any month during the monsoon season.

Park et al. (2008) report an $O_3$ minimum inside the ASMA based on retrievals from another space borne sensor (ACE-FTS). They flag profiles as "inside" the ASMA, if $CO \geq 60$ nmol/mol at 16.5 km. Co-located retrievals then show decreased mixing ratios of $O_3$ and stratospheric tracers (e.g. HCl) for the "inside" bin, as compared to

15 the "outside" bin. This approach rather identifies an anti-correlation of CO and $O_3$ in the UT than generally decreased $O_3$ inside the ASMA. Such an anti-correlation is also indicated in the EMAC simulated snapshot shown in Fig. 2.

Summarizing, our simulation is able to reproduce decreased $O_3$ in the ASMA for those special circumstances it has been reported by other studies. At the same time, we found no indication in our simulation to expect

decreased $O_3$ in the ASMA at about 150 hPa in September, although only two analyzed seasons strictly may provide not more than a strong indication of climatological trace gas signatures in the ASMA. Possible reasons for the differences between July/August versus September might include the longer time available for photochemical buildup of $O_3$ in the ASMA, decreasing resupply of $O_3$-poor air towards the end of the monsoon season, more inmixing of $O_3$-rich TL air into the decaying ASMA (dynamical instabilities), and changes to the

altitudes of maximum $O_3$ production.

## 8    Summary

Our study contributes to the so far sparse ASMA in-situ measurements, allowing us to address some of the aspects of this important UT phenomenon that were recently identified as poorly understood (Randel et al.,

2016): dynamical and chemical coupling with convection, composition/reactive chemistry in the monsoon region, and mixing of higher latitude lower stratospheric air into the tropical TL by the ASMA.

Data from the HALO ESMVal campaign that were gathered during a flight from Male (Maledives) to Larnaca (Cyprus) on 18 September 2012 are presented and analyzed. That region is particularly unexplored by in-situ measurements. HYSPLIT backward trajectories show that HALO most of the time was in an UT filament, which

had been part of the UT ASMA circulation for at least 10 days, thereby circulating around the anticyclone. Uplifted air was entrained into the UT filament at the eastern ASMA flank, which was then transported by





almost unperturbed, parallel streamlines in the southern ASMA fringe (Fig. 3). Back-trajectories indicate that HALO crossed the filament three times in the zone where an originally larger ASMA was just splitting into an Iranian and a Tibetan part (Fig. 4). At least a part of the filament from the eastern ASMA flank is diverted into the new Iranian anticyclone. From the in-situ data available, the first transect of the fringe filament provides the

hitherto most direct view of the upstream eastern ASMA flank, where several processes act that have the potential to strongly modify UT trace gas mixing ratios.

A global simulation with the EMAC model matches observed trace gas mixing ratios along the HALO flight track reasonably well. The specified dynamics setup (nudging) certainly enforces a better match between simulation and observations, compared to what could be expected from a free-running simulation. The synoptic

scale of the ASMA acts to alleviate discrepancies that are related to the limited spatial and temporal resolution of the simulation, but a perfect match cannot be expected. Overall we find that this simulation is well suited to be used for further interpretation of the measurements. An ASMA splitting event indicated by back-trajectories is also reproduced by the EMAC simulation and further analyzed in the accompanying paper.

Based on the general picture provided by previous studies, depleted $O_3$ mixing ratios were expected in the

ASMA compared to the surrounding UT. However, enhanced $O_3$ was found in the ASMA filament encountered by HALO. In order to identify the processes that generated this $O_3$ signature, additional tracers are considered for further analyses: CO as marker for lower tropospheric air, HCl for stratospheric or TL origins, NO and $NO_y$ as important players in $O_3$ photochemistry. All above mentioned tracers were measured in-situ, and their mixing ratios steeply increase across the boundary of the ASMA filament compared to the adjoining clean air in the

south.

Tracer-tracer relations of the in-situ data are consistent with a mix of UT and lower tropospheric air in the ASMA fringe. Two effects likely have contributed to the observed signatures of increased $O_3$: photochemical $O_3$ production and entrainment of stratospheric or TL air. The EMAC simulation indicates that net photochemical $O_3$ production is maximal, where uplifted air with $O_3$ precursors originating in boundary layer pollution (e.g.

CO) mixes with UT air that is enriched in (lightning) NO, another precursor. Besides of increased $O_3$, mixing ratios of the stratospheric tracer HCl are also relatively enhanced in air that had been part of the UT ASMA for longer. This trace gas signature can not be explained by photochemical ageing of uplifted, lower tropospheric air alone. The EMAC simulation indeed shows that a TL filament with more stratospheric trace gas signatures than the surrounding UT air is entrained into the ASMA fringe at a tropopause through at the eastern flank of the

anticyclone (Fig. 2, 6). It is dragged away from the TP and deeper into the troposphere, circling around the ASMA interior. That particular event did hardly contribute to the simulated data on the flight track, but timing and location are such that – given the uncertainties of the simulation - the corresponding event in reality might still have contributed to the observed air composition. If not this, then earlier such entrainment events contributed to the ASMA trace gas signatures – in both, simulation and measurements.

Dynamical instabilities, like the ASMA splitting event encountered by HALO, provide a means to overcome the radial transport barriers presented by closed streamlines, and to effectively stir the entrained air into the ASMA interior.



Our current study focuses on the detailed analysis of a single transect of in-situ data through one part of the ASMA, close to the end of the monsoon season. The relevance of entrainment of TL air into the ASMA fringe, photochemistry and stirring in the ASMA for the trace gas budgets of the ASMA is further explored in the accompanying paper. However, here we also found that the EMAC simulation is able to reproduce decreased $O_3$

mixing ratios in the ASMA at 100 hPa for July and August as reported by previous studies, but it also reproduces increased $O_3$ as observed during the HALO ESMVal campaign. No decreased $O_3$ was found in the simulation for lower altitudes or September monthly mean values, and the apparent contradiction to previous studies vanishes in this more differentiated view. The incidence of $O_3$-rich air in the ASMA –as seen in the simulated monthly mean data- indicates that the ESMVal in-situ measurements could even represent a common

composition of the ASMA at about 150 hPa.

### Author contributions

K. Gottschaldt analyzed the EMAC and in-situ data, conducted the Lagrangian calculations, produced the plots and drafted the paper. H. Schlager conceived the study, led the ESMVal HALO campaign, interpreted EMAC

and in-situ data. R. Baumann wrote and helped with the code that facilitated the HYSPLIT calculations. V. Eyring conceived and led the ESMVal project. H. Bozem and P. Hoor supplied the CO in-situ measurements. P. Jöckel led the ESCiMo project, coordinated the preparation of and conducted the EMAC simulations. T. Jurkat and C. Voigt supplied the HCl measurements. A. Zahn was responsible for the $O_3$ measurements. H. Ziereis contributed the $NO_y$ measurements. All authors contributed to the text.

### Acknowledgements

The authors gratefully thank H. Garny for valuable comments on the manuscript, B. Brötz, F. Frank, K. Graf, V. Grewe, H. Huntrieser, P. Konopka, R. Müller, M. Nützel, L. Pan, R. Ren, and B. Vogel for helpful discussions.

We thank the German Science Foundation DFG for funding within HALO-SPP 1294 under contracts JU 3059/1-

1, SCHL 1857/2-2, SCHL 1857/4-1, VO 1504/2-1 and VO 1504/4-1. The ESMVal aircraft campaign was funded by the DLR-Project ESMVal. KG and HS appreciate support by the EU project StratoClim (grant no. 603557) and BMBF project Spitfire (grant no. 01LG1205B. CV and TJ thank financing by the Helmholtz Association under contract no. VH-NG-309 and under contract No. W2/W3-60. In addition we thank the flight department of DLR for their great support during the campaign. S. Müller contributed to the CO measurements and S.

Kaufmann supervised the HCl measurements during the flight.

The EMAC model simulations were performed at the German Climate Computing Centre (DKRZ) through support from the Bundesministerium für Bildung und Forschung (BMBF). DKRZ and its scientific steering committee are gratefully acknowledged for providing the HPC and data archiving resources for the projects 853 (ESCiMo - Earth System Chemistry integrated Modelling) and 854 (ESMVal - Erdsystemmodellevaluierung).



We used the NCAR Command Language (NCL) for data analysis and to create some of the figures of this study. NCL is developed by UCAR/NCAR/CISL/TDD and available on-line: http://dx.doi.org/10.5065/D6WD3XH5.

The article processing charges for this open-access publication were covered by a Research Centre of the Helmholtz Association.

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




|  | Classification | UTC | Remarks | Fig. |
|---|---|---|---|---|
| POI1 | Ascent from Male + UT south of ASMA | 05:36 06:40 | Clean air, not related to ASMA | - |
| POI2 | Southern ASMA boundary region | 06:40 07:46 | Back-trajectories related to ASMA, but clean air dominates | S4 |
| POI3 | Outer ASMA streamlines | 07:46 08:21 | UT ASMA circulation + air uplifted at eastern flank; 3 days after passing eastern flank | 3 |
| POI4 | Dive over Arabic Peninsula | 08:21 09:05 | lower ASMA boundary (~180 hPa) to 650 hPa | S4 |
| POI5 | Outer ASMA streamlines | 09:05 10:50 | As POI3, but filament curled in; 6 days after passing eastern flank | 3 |
| POI6 | Outer ASMA streamlines | 10:50 11:53 | As POI3, but less uplifted air; 5 days after passing eastern flank | 3 |
| POI7 | Descend to Larnaca | 11:52 12:29 | Lower ASMA boundary to ground | S4 |

**Table 1: Periods of interest for the measurements during the HALO ESMVal flight from Male to Larnaca on 18 September 2012. Column UTC shows the time periods of the measurements.**



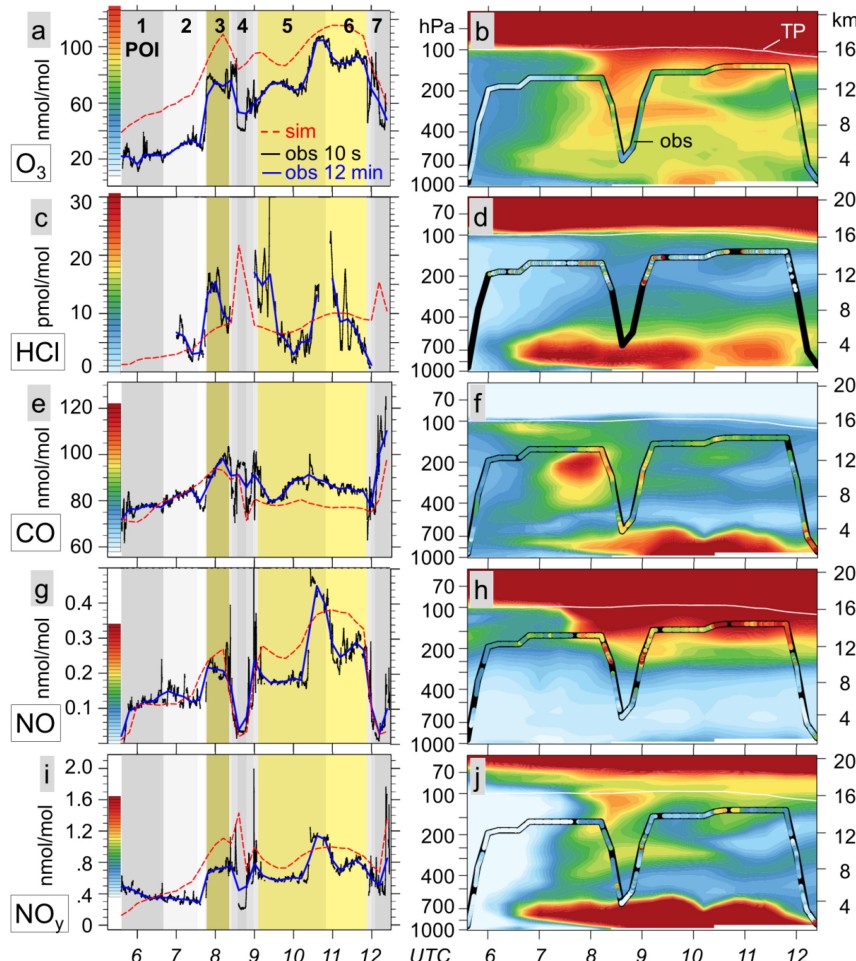

**Figure 1:** The left panel shows mixing ratios of $O_3$, HCl, CO, NO and $NO_y$ along the HALO flight track from Male to Larnaca, on 18 September 2012. Black: in-situ measurements in 10 s resolution, Blue: interval averages of the in-situ data, corresponding to 12 min simulation time steps, Red: simulation results. Yellow shadings mark the periods of interest, see text. Corresponding curtains simulated with EMAC along the flight track are shown on the right. The pipe follows the HALO flight altitude, filled with measured trace gas mixing ratios in the same colour coding (legends integrated in corresponding left panels). Black = missing. A thin white line indicates the simulated tropopause.





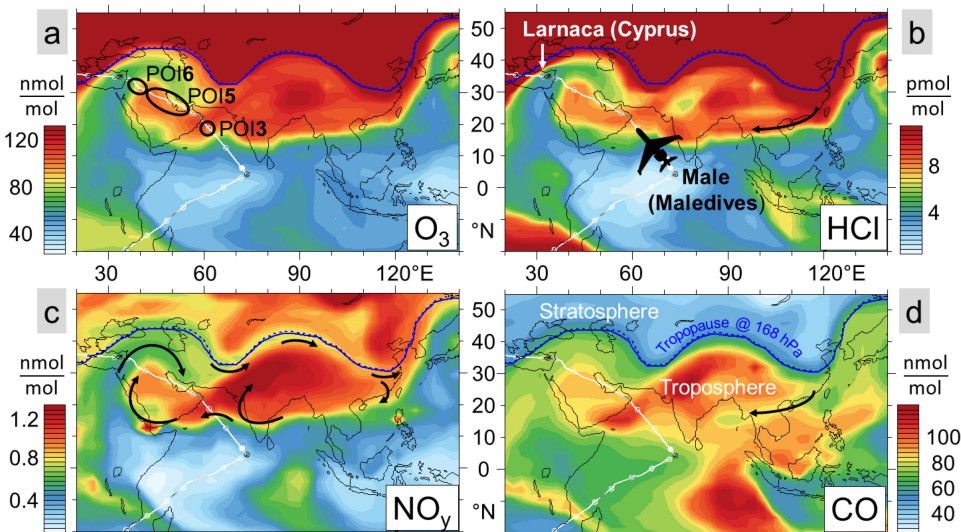

**Figure 2: Mixing ratios of O$_3$, HCl, NO$_y$ and CO, as simulated by EMAC for 6 UTC on 18 September 2012, at 168 hPa. Beads along the flight track are separated by 1 hour, and the tail end of the HALO silhouette marks the actual position. Black circles in panel (a) indicate the HALO position during the periods of interest, which represent the ASMA measurements. During POI3 HALO was flying in an altitude range containing the shown pressure level. Arrows illustrate the wind field (panel c) and also highlight a filament of TP layer entrainment into the free troposphere at the eastern ASMA flank, indicated by anti-correlated mixing ratios of HCl and CO (panels b and d).**



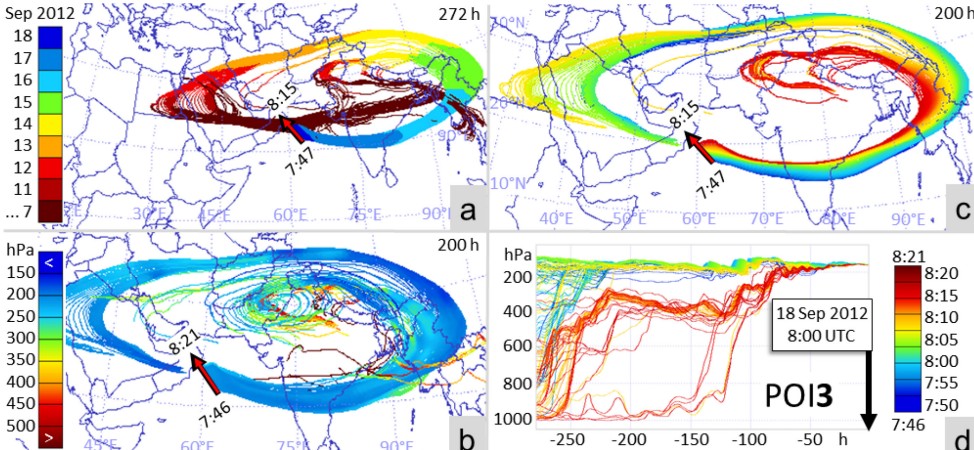

**Figure 3: HYSPLIT-simulated backward trajectories starting at the HALO flight track (red arrows). The integration length is noted in the upper right corner of each lat-lon panel. All trajectories of one panel start at the same time, approximately in the middle of the period corresponding to the respective flight segments (panels d, h, m). Note that panel a covers a slightly longer flight segment than panels b and c, which is needed to discuss the delimitation of POI3. Colors in panels a, e, g relate the previous positions of the measured air parcels to calendar days. Colors in b, f, j show pressure altitude, which is supplemented by altitude vs time in d, h, m. Individual trajectories are color coded according to the time of measurement at their respective starting positions in panels c, d, g, h, k, m. Panels a, f, i show the same map section, while all others are scaled individually.**





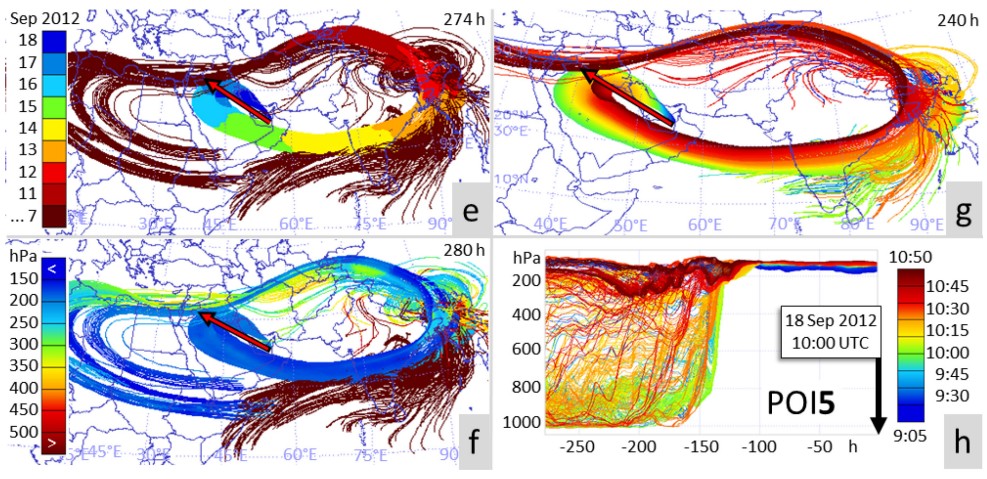

**Figure 3:** Continued for POI5.

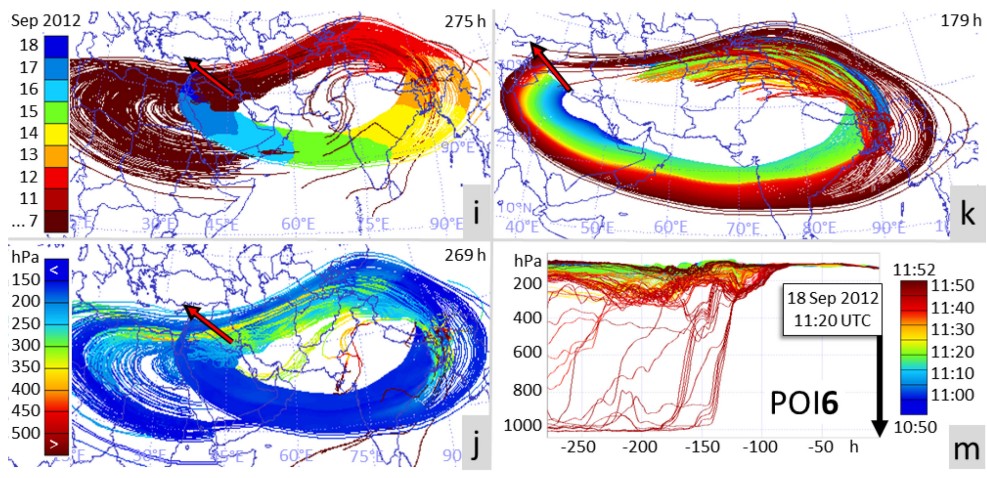

**Figure 3:** Continued for POI6.





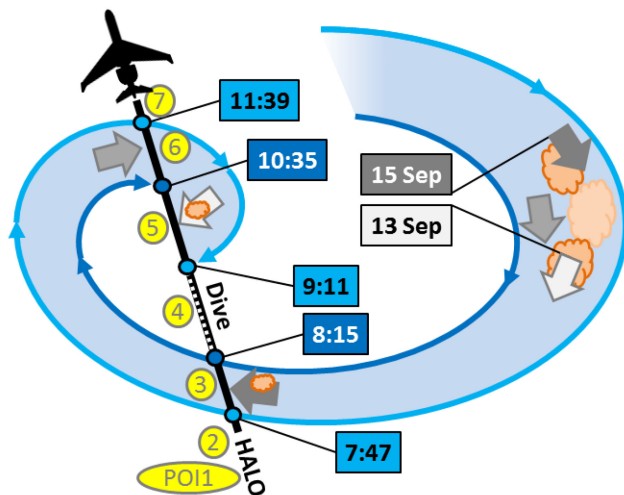

**Figure 4: Schematic of the filament of UT ASMA air that was transected by HALO during the ESMVal campaign. The encountered air parcels all had a similar genesis: UT air travelling in the ASMA fringe was to different degrees entrained by deep convection at the eastern ASMA flank, then continuing along the southern flank of the anticyclone to the respective measurement locations. POI3 provides the most direct view of the eastern ASMA flank. The schematic is in ASMA centered coordinates, with relative positions of air masses (grey arrows) and convection (orange clouds) indicated. The interior trajectory of the filament and corresponding times of measurement are indicated by dark blue and the exterior trajectory is light blue. Grey shades and white indicate the air masses encountered during the respective three periods of interest (POI3/5/6), and to the corresponding date they passed the eastern flank.**





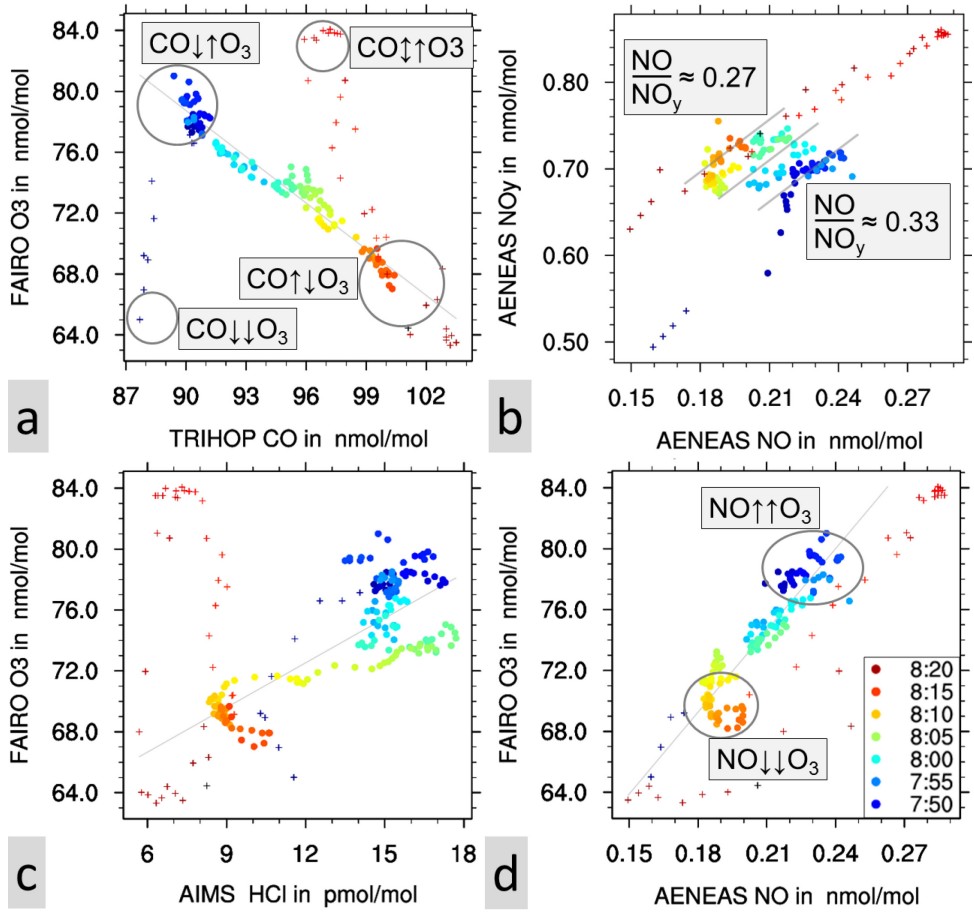

**Figure 5: Tracer-tracer relations, as observed during one transect through the ASMA fringe. Colors indicate the time of measurement, filled circles highlight the first period of interest (POI3), grey lines are linear fits to those data (panels a, c, d) or hand-drawn markings discussed in the text (panel b), and crosses show data just before and after POI3. Reservoirs referred to in the text are marked by grey boxes.**





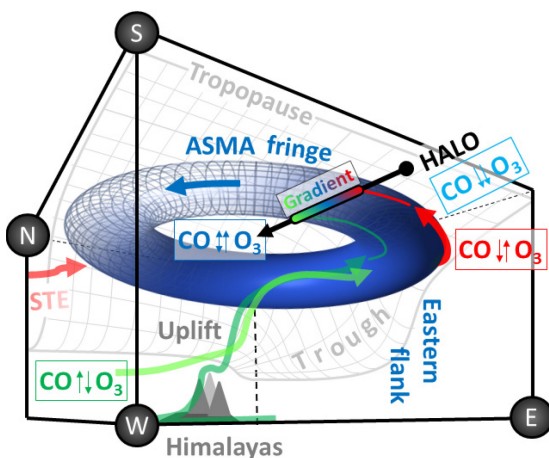

**Figure 6: Schematic of the synoptic situation and CO-O$_3$ trace gas signatures that contributed to the HALO in-situ measurements transecting the southern ASMA fringe during the ESMVal campaign (POI3). The foreshortened cuboid approximately covers 15°N – 40°N, 40°E – 120°E, and surface to 100 hPa, with features not drawn to scale. Recent contributions of stratospheric intrusions ("STE") were not detected in the measurements, but might have contributed to the fringe's trace gas signatures earlier. See text for details.**



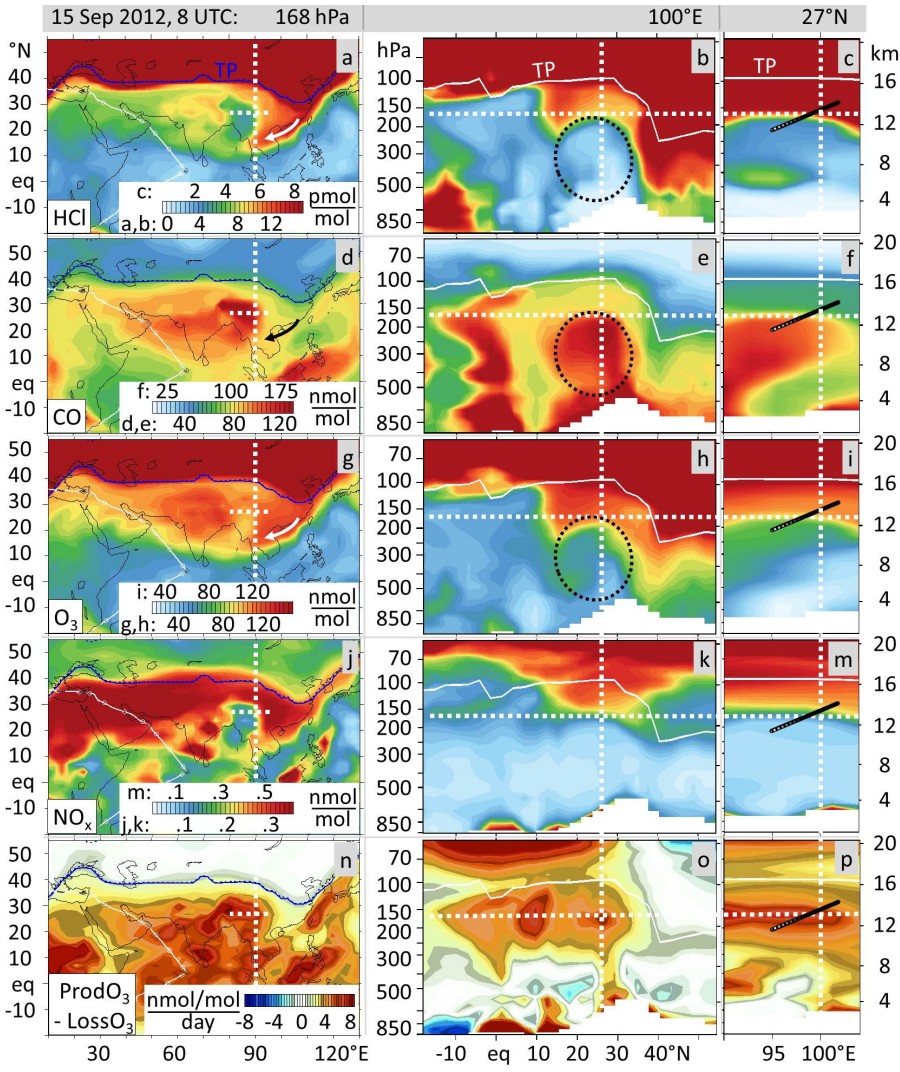

**Figure 7.** Mixing ratios of HCl, CO, $O_3$, $NO_x$, and net photochemical $O_3$ production as simulated by EMAC for 8 UTC on 15 September 2012. Black bars in the right column indicate the air mass to be encountered by HALO three days later. That air mass was passing the eastern flank of the anticyclone at the pictured time. Its eastern part was entrained by uplifted air there, creating a trace gas gradient in the fringe. Zonal and meridional curtains are shown on the right, and positions of the two respective other panels are marked by dashed lines in each panel. Arrows and circles indicate features discussed in the text.