# Peer review of "Trace gas composition in the Asian summer monsoon anticyclone: A case study based on aircraft observations and model simulations"

_Atmospheric Chemistry and Physics, 2016_

## Referee Comment (RC1) · Anonymous Referee #1 · 22 Dec 2016

This manuscript presents new perspective on transport in the Asian summer monsoon anticyclone based on newly obtained in-situ measurements, Lagrangian trajectory model calculations and also simulations from a global chemistry transport model. Tropospheric and stratospheric tracers, such as, CO, NOy, O3, and HCl are used in the analyses throughout the study. The careful and detailed analyses on both the in-situ measurements and model simulations provide valuable information on large scale transport in the Asian summer monsoon anticyclone in the upper troposphere region focused on measurements during a field campaign conducted in September 2012. I would suggest considering making this paper a little more concise so that the focus of the findings can be emphasized.

General Comments

- It seems crucial for this study to define the anticyclonic boundaries explicitly. There is no clear indication of when the anticyclonic circulation was strong, in terms of temporal and spatial variability throughout this study.

- POI 1 through 7 – Dividing the flight tracks into multiple segments (POIs) provide great detail on the dynamical and chemical evolution of the anticyclonic circulation. However, it seems that there is a lot of information to digest. One thing the authors might consider is focusing on the relevant flight segments that is more relevant to this study to emphasize all the findings of this study.

Specific Comments

P2, L5-6 (Abstract) - This statement is somewhat misleading. Previous studies may have shown ozone minimum inside the anticyclone during monsoon maximum period (July-August) and this study is focused on measurements in September near the edge of the anticyclone. I would recommend removing this sentence to avoid any confusion.

P2, L26 - What is 'above processes' referring to? It would be better to describe it explicitly. Also it is important to mention this process is important for trace gas budgets in September.

P3, L14 and 27 (Introduction) - June to September (reference). - South of the anticyclone (reference).

P3, L29 - Konopka et al., 2010 (http://www.atmos-chem-phys.net/10/121/2010/acp-10-121-2010.pdf) can be a good reference here. For instance, their Fig. 7 shows general idea of transport of ozone in the Asian monsoon anticyclone.

P4, L22-23 - EMAC global simulations. . .reproduced well -> EMAC global model. . .reproduced well in the simulations.

P5, L8, 12 & 15 (section 2) - Full names of GLORIA, ANEAS and FAIRO have to be

included here.

P5, L26-27 - the aircraft campaign – Is this referring to one specific campaign or multiple campaigns in general?

P6, L4-5 - References for those tropopause definitions should be included here.

P6, L14 - represented by this approach indirectly -> represented indirectly

P6, L22 (section 3) - sudden increase -> It may help to add exact flight time (for example: 7:50 am, UTC).

P6, L26 - yellowish -> yellow-brown to yellow

P6, L29-30 - positions of streamlines -> For better definition of the interior vs. fringe of the anticyclone, it might be helpful to add a figure showing streamlines (or boundaries) of the anticyclone. This can also be done by adding streamlines (or boundaries) in Fig. 2.

P6, L35 - (supplementary material) -> (supplementary material, S4)

P8, L11 - Geographical locations, such as, Larnaca and Oman can be marked on the map if it is necessary.

P8, L17-18 - Deep convection. . .considerably to POI5 -> I assume this statement refers backward trajectories from the surface reaching up to higher altitude as deep convection. I am wondering if there is any other evidence of showing deep convective activities during POI5.

P8, L31-33 - Highly polluted. . .though. -> This sentence is vague. It is not clear how this effect is shown in POI4.

P9, L2 - measurement locations. -> Is this based on the trajectory calculations?

P9, L8 - more natural definition -> meaning of this is not clear.

P10, L5 (section 4) - stratospheric or TL air then -> stratospheric or TL air

P10, L23-24 - by the coarser output (time) resolution directly, and indirectly because the representation of processes is limited by the grid resolution -> because of the coarser temporal resolution of the output (direct) and the representation of processes is limited by the grid resolution (indirect).

P11, L2 - synoptic scale feature -> I am not sure I agree with this statement that ASMA is synoptic scale feature. ASMA itself is a global scale feature and its variability is represented in this study.

P11, L15 (section 5) - NO(x) -> NOx

P12, L25, 27 - then or though at the end of the sentences might not be necessary

P14, L1 (section 6) - I wonder if there is a way to reduce the amount of discussions in section 6.1. This section only serves as an introduction and contains very detailed discussion.

P14, L15 - Konopka et al. (2010) can be mentioned here.

P15, L2 - formed -> formed

P15, L7-8 - artefact of. . .components -> I am not sure what the meaning of this sentence is.

P15, L23 - O3 correlates with HCl -> O3 is positively correlated with HCl

P16, L8 - where. . .during POI3 -> where measurements of O3 were lower than outer streamlines during POI3.

P16, L10 - minted?

P16, L11 - Contrary -> On the contrary

P17, L18-25 - This paragraph can be revised just to emphasize that the focus of this study is on the analyses based on measurements during September (not Jul-Aug) and in the fringe of the anticyclone (compared to inside).

P17, L30 (section 8) - I am not convinced that this study has showed dynamical and chemical coupling with convection.

P18, L4 - From…available -> Based on the in-situ measurements data,

P18, L7-13 - match -> agreement

P18, L30 - Fig. 2, 6 -> Figs. 2 and 6

Figures (Supplement) - What do colors in Figs. 4S (c, g, k) mean? Maybe the same as d, h, m?

---

## Referee Comment (RC2) · Anonymous Referee #2 · 4 Jan 2017

**General:**

The paper shows new in-situ data observed in the upper tropospheric part of the Asian monsoon anticyclone. Because trace gas observations in this region are very rare, it is important to publish this data. The observations show a very unique type of air composition in this region with contributions of variety of different processes. Due to the nature of the subject, it is probably difficult to give a more clear picture. As one of the most important findings, the unexpected high ozone levels are reported. Of course, it is difficult to explain this effect only from experimental data. However, using model (EMAC), it would be desirable to see how in the region around 150 hPa, the forming of the enhanced ozone levels during summer and September can be understood, or,

at least is resolved by EMAC. Furthermore, the paper tries to report everything one can say without trying to concentrate on the most important points. Thus, I would recommend to get rid of some unnecessary ballast (for some ideas see below). Thus because of these two reason (missing model-related explanation of enhanced ozone levels and too many details) the paper needs a major revision.

**Minor points:**

1. P 3/L 25
   "On the contrary" - what do you mean. Contrary to "no decrease" is decrease. Please reformulate

2. P 6/L 27
   You are talking about streamlines but your never show them. You are only showing trajectories which, in general, do not follow streamlines. (only for a stationary flow streamlines and trajectories are the same lines). After presenting your data (Fig 1) and the vertical and horizontal cross sections through the model (Fig 2) it would be nice to see also the meteorology at e.g. p=150 hPa showing streamlines of the geopotential for few days before the flight.

3. P 6/L 33
   POI2 - here is potential to shorten the text. This flight segment has nothing to do with ASMA

4. P 7/L 15
   POI4 - too much information. However, the difference between the slow rotation (lower level) and fast rotation (higher level) is an interesting feature.

5. P 7/P 8
   POI5/POI6 - in my opinion there is no reason to discuss these two flight segments separately. Also one trajectory figure would be enough. The difference between

the Iranian and Tibetan mode cannot be seen from your investigation. I would recommend to remove this part of the text

6. caption of Fig 3

Please explain only a, b, c, and d panels in this caption. For POI5/6 would be enough to write: same like for POI3

7. P9-11

I agree that the model performs good to represent the in-situ measurements. In the following chapter the tracer-tracer correlations are discussed. It would be nice to see (or only to know) how such correlations are represented in the model. Typically, models do not correctly represent such correlations. Maybe EMAC is better?

8. P11/L3

"...which is carried forward to the related large scale trace gas distributions" - this sentence is not clear for me. Please reformulate

9. P11 L15

"might not leave too much freedom" - much too speculative

10. P13 L20

Figure 6 is difficult to understand. In particular the marks "N" and "S" are very confusing. I do not see north or south of the ASMA here.

---

## Author Comment (AC1) · 8 Mar 2017

**Reply to Anonymous Referee #1**

The authors would like to thank the reviewer for those well-founded comments, which helped to increase the readability and the scientific quality of the paper.

General comments:

*It seems crucial for this study to define the anticyclonic boundaries explicitly. There is no clear indication of when the anticyclonic circulation was strong, in terms of temporal and spatial variability throughout this study.*

Placing the measurements into the context of temporal and spatial ASMA variability was already part of an earlier version of the paper, but a thorough discussion of that aspect is very voluminous and we could not reconcile it with a detailed discussion of the measurements in one concise paper. Now the more climatological aspects are the focus of an accompanying follow-up study (Gottschaldt et al., 2017). Nevertheless we agree with the reviewer that the evolution of the ASMA before the flight is relevant for understanding the measurements. We added a new figure (now Fig. 3) to the main text, showing a daily sequence of streamlines and geopotential height from 9 days before to 1 day after the HALO flight at the altitude of the measurements. Another figure (now S1) has been added to the supplement, showing streamlines over PV on an isentropic level. Geopotential and PV were proposed as proxies for characterizing the ASMA and its boundaries (Barret et al., 2016; Ploeger et al., 2015), alternatively on pressure or isentropic coordinates.
Streamlines represent an instantaneous snapshot of transport barriers, because there is no large scale transport perpendicular to streamlines. This complements the existing trajectory plots, which in contrast to streamlines include information about the evolution of the time dependent flow.
The ASMA boundary definitions we are aware of (Ploeger et al., 2015; Barret et al., 2016; Pan et al., 2016) are based on Eulerian fields (PV, geopotential, wind) and emphasize the concept of a closed ASMA volume or transport barriers on monthly or seasonal time scales. We refrain from using such a methodology in the study, because our analyses rather aim to explain observed UT tracer distributions resulting from daily-scale dynamics. The latter is best captured by Lagrangian trajectories, which unlike the above Eulerian approaches inherently reflect the time-dependence of the flow. Our choice of POIs and the delineation of the ASMA is thus based on back-trajectories (starting from the flight path) and the observed trace gas signatures. Some elaboration on our approach has been added to section 3.

*POI 1 through 7 – Dividing the flight tracks into multiple segments (POIs) provide great detail on the dynamical and chemical evolution of the anticyclonic circulation. However, it seems that there is a lot of information to digest. One thing the authors might consider is focusing on the relevant flight segments that is more relevant to this study to emphasize all the findings of this study.*

Documenting the HALO ESMVal measurements in the ASMA is one of the objectives of this paper, also as a base for follow-up studies. Nevertheless we agree with the reviewer that the paper would be more concise, if not all flight segments are discussed in great detail. Considering the recommendations of reviewer 2 we even went a step further and now consequently focus on POI3 in the main text. All other flight segments have been moved to an appendix. They are now discussed in the main text only as far as needed to put POI3 into the context of the entire flight.

Specific comments:

*P2, L5-6 (Abstract) - This statement is somewhat misleading. Previous studies may have shown ozone minimum inside the anticyclone during monsoon maximum period (July-August) and this study is focused on measurements in September near the edge of the anticyclone. I would recommend removing this sentence to avoid any confusion.*

Reformulated, avoiding the reference to previous studies.

*P2, L26 - What is 'above processes' referring to? It would be better to describe it explicitly. Also it is important to mention this process is important for trace gas budgets in September.*

In the revised version the processes are stated explicitly and it is pointed out that trace gas signatures differ between July/August and September.

*P3, L14 (Introduction) - June to September (reference).*

We included references for the interpretation of the monsoon as sea breeze and for the monsoon period.

*P3, L27 (Introduction) - South of the anticyclone (reference).*

We reformulated the sentence to make clear that we refer here to our measurements only. The location of the anticyclone at the time of the flight arises from streamlines, back-trajectories and the definition of POI3 in our paper.

*P3, L29 - Konopka et al., 2010 (http://www.atmos-chem-phys.net/10/121/2010/acp-10-121-2010.pdf) can be a good reference here. For instance, their Fig. 7 shows general idea of transport of ozone in the Asian monsoon anticyclone.*

Konopka et al. (2010) had not been mentioned in the original draft, because they focus on the stratosphere, and we focus on the troposphere. However, we agree with the reviewer that their idea of ozone transport is related to the in-mixing we are describing, and we added a corresponding reference to section 1.

*P4, L22-23 - EMAC global simulations ... reproduced well -> EMAC global model ... reproduced well in the simulations.*

According to the EMAC glossary at www.messy-interface.org, "model" refers to one executable. We only refer to a specific set of simulations with a specific model setup (chemistry climate model with specified dynamics) in the paper. We prefer to stick with the original formulation, because there are other simulations with the same executable, but different model setups.

*P5, L8, 12 & 15 (section 2) - Full names of GLORIA, ANEAS and FAIRO have to be included here*

Done.

*P5, L26-27 - the aircraft campaign – Is this referring to one specific campaign or multiple campaigns in general?*

The specified dynamics setup is designed to globally reproduce observed large scale dynamics. It is understood that the results need to be validated for each region. Output of the Eulerian EMAC model is

available globally on a Gaussian grid and should in general be suited for all campaigns falling into the simulated period. A corresponding statement has been added to the text.

*P6, L4-5 - References for those tropopause definitions should be included here.*

Done.

*P6, L14 - represented by this approach indirectly -> represented indirectly*

Done.

*P6, L22 (section 3) - sudden increase -> It may help to add exact flight time (for example: 7:50 am, UTC).*

Done.

*P6, L26 - yellowish -> yellow-brown to yellow*

It's called "shades of yellow" in the revised version.

*P6, L29-30 - positions of streamlines -> For better definition of the interior vs. fringe of the anticyclone, it might be helpful to add a figure showing streamlines (or boundaries) of the anticyclone. This can also be done by adding streamlines (or boundaries) in Fig. 2.*

A sequence of streamline plots has been added as Fig. 3 (see also the first section under "General comments").

*P6, L35 - (supplementary material) -> (supplementary material, S4)*

Done.

*P8, L11 - Geographical locations, such as, Larnaca and Oman can be marked on the map if it is necessary.*

Larnaca and Male were already marked in Fig. 2, and Oman has been added. Adding more geographical locations might not be necessary, because in the revised version there is less emphasis on the discussion of the entire flight.

*P8, L17-18 - Deep convection … considerably to POI5 -> I assume this statement refers backward trajectories from the surface reaching up to higher altitude as deep convection. I am wondering if there is any other evidence of showing deep convective activities during POI5.*

Yes, there is. Fig. A4d (please see the look-up table at the end of this document for a mapping between figures in the original and the revised manuscript) shows that uplift took place at and before about 13 September 2012, 0 UTC. Fig. A4b shows that the uplift was confined to the eastern ASMA flank. This location has now been marked in the corresponding Meteosat pictures of Fig. S8 (orange circles). White shades dominate within the circles, indicating convective clouds. Fig. S8 had to be extended backwards in time to cover the discussed period.

*P8, L31-33 - Highly polluted … though. -> This sentence is vague. It is not clear how this effect is shown in POI4.*

We removed the sentence.

*P9, L2 - measurement locations. -> Is this based on the trajectory calculations?*

Yes. We included a reference to Figs. 4a / A4a / A5a (originally Fig. 3aei) here to make this clear.

*P9, L8 - more natural definition -> meaning of this is not clear.*

Formulation changed to "… more information about …".

*P10, L5 (section 4) - stratospheric or TL air then -> stratospheric or TL air*

Done.

*P10, L23-24 - by the coarser output (time) resolution directly, and indirectly because the representation of processes is limited by the grid resolution -> because of the coarser temporal resolution of the output (direct) and the representation of processes is limited by the grid resolution (indirect).*

Done.

*P11, L2 - synoptic scale feature -> I am not sure I agree with this statement that ASMA is synoptic scale feature. ASMA itself is a global scale feature and its variability is represented in this study.*

We changed the formulation to "large scale feature" to avoid any confusion.

*P11, L15 (section 5) - NO(x) -> NOx*

Our notation was intended to cover discussions about NO (measured and modelled) and NOx (modelled only). We changed it to "NO vs O3, NOx vs O3".

*P12, L25, 27 - then or though at the end of the sentences might not be necessary*

Deleted.

*P14, L1 (section 6) - I wonder if there is a way to reduce the amount of discussions in section 6.1. This section only serves as an introduction and contains very detailed discussion.*

In the revised version this section is more focused on our case. Some of the discussion of previous work has been moved to section 1.

*P14, L15 - Konopka et al. (2010) can be mentioned here.*

We agree with the reviewer. However, in response to the previous comment we decided to reduce the discussion of previous work in section 6.1., and Konopka et al. (2010) has been added to section 1 instead.

*P15, L2 - foormed -> formed*

Done.

*P15, L7-8 - artefact of … components -> I am not sure what the meaning of this sentence is.*

Reformulated to: "Ascending trajectories in that region could also be the result of convective activity that is not explicitly resolved in the reanalysis data, but still represented as regional uplift."

*P15, L23 - O3 correlates with HCl -> O3 is positively correlated with HCl*

Done.

*P16, L8 – where … during POI3 -> where measurements of O3 were lower than outer streamlines during POI3.*

Done.

*P16, L10 - minted?*

Minted → reformulated

*P16, L11 - Contrary -> On the contrary*

Done.

*P17, L18-25 - This paragraph can be revised just to emphasize that the focus of this study is on the analyses based on measurements during September (not Jul-Aug) and in the fringe of the anticyclone (compared to inside).*

The paragraph has been reformulated to better relate to the problem posed at the beginning of the chapter, and also to the rest of the paper.

*P17, L30 (section 8) - I am not convinced that this study has showed dynamical and chemical coupling with convection.*

That bullet point of Randel et al. is included here, because the interplay of convective uplift and subsequent transport in the ASMA are discussed in connection with tell-tale trace gas signatures to explain the measurements.

*P18, L4 – From … available -> Based on the in-situ measurements data,*

Done.

*P18, L7-13 - match -> agreement*

Done.

*P18, L30 - Fig. 2, 6 -> Figs. 2 and 6*

Done.

*Figures (Supplement) - What do colors in Figs. 4S (c, g, k) mean? Maybe the same as d, h, m?*

These figures are now part of Appendix A and the captions have been clarified.

**References**

Barret, B., Sauvage, B., Bennouna, Y., and Le Flochmoen, E.: Upper-tropospheric CO and O3 budget during the Asian summer monsoon, Atmospheric Chemistry and Physics, 16, 9129-9147, 10.5194/acp-16-9129-2016, 2016.

Gottschaldt, K., Schlager, H., Baumann, R., Cai, D. S., Eyring, V., Graf, P., Grewe, V., Hoor, P., Jöckel, P., Jurkat, T., Voigt, C., Zahn, A., and Ziereis, H.: Working title: Interplay of dynamics and composition in the Asian summer monsoon anticyclone, Atmos. Chem. Phys. Discuss., in prep., 2017.

Konopka, P., Grooß, J.-U., Günther, G., Ploeger, F., Pommrich, R., Müller, R., and Livesey, N.: Annual cycle of ozone at and above the tropical tropopause: observations versus simulations with the Chemical Lagrangian Model of the Stratosphere (CLaMS), Atmos. Chem. Phys., 10, 121-132, 10.5194/acp-10-121-2010, 2010.

Pan, L. L., Honomichl, S. B., Kinnison, D., Abalos, M., Randel, W. J., Bergman, J. W., and Bian, J.: Transport of chemical tracers from the boundary layer to stratosphere associated with the dynamics of the Asian summer monsoon, Journal of Geophysical Research: Atmospheres, 121, 10.1002/2016JD025616, 2016.

Ploeger, F., Gottschling, C., Griessbach, S., Grooß, J. U., Guenther, G., Konopka, P., Müller, R., Riese, M., Stroh, F., Tao, M., Ungermann, J., Vogel, B., and von Hobe, M.: A potential vorticity-based determination of the transport barrier in the Asian summer monsoon anticyclone, Atmospheric Chemistry and Physics, 15, 13145-13159, 10.5194/acp-15-13145-2015, 2015.

| Original manuscript | Revised version |
|---|---|
| Fig. 1 | Fig. 1 |
| Fig. 2 | Fig. 2 |
|  | Fig. 3 |
| Fig. 3 abcd | Fig. 4 abcd |
| Fig. 4 | Fig. 5 |
| Fig. 5 | Fig. 6 |
| Fig. 6 | Fig. 7 |
| Fig. 7 | Fig. 8 |
| Fig. S4 abcd | Fig. A1 |
| Fig. S6 | Fig. A2 |
| Fig. S4 efgh | Fig. A3 |
| Fig. 3 efgh | Fig. A4 abcd |
| Fig. 3 ijkm | Fig. A5 abcd |
| Fig. S4 ijkm | Fig. A6 abcd |
|  | Fig. S1 |
| Fig. S1 | Fig. S2 |
| Fig. S2 | Fig. S3 |
| Fig. S3 | Fig. S4 |
| Fig. S5 | Fig. S5 |
| Fig. S7 | Fig. S6 |
| Fig. S8 | Fig. S7 |
| Fig. S9 | Fig. S8 |

Table 1: Mapping of figures between original and revised manuscript

---

## Author Comment (AC2) · 8 Mar 2017

**Reply to Anonymous Referee #2**

The authors would like to thank the reviewer for her/his ideas, which helped a lot to make the paper more concise. We also appreciate the insightful comments on desirable additional analyses, which match the scope of an almost finished follow-up study.

General comments:

*"The paper shows new in-situ data observed in the upper tropospheric part of the Asian monsoon anticyclone. Because trace gas observations in this region are very rare, it is important to publish this data. The observations show a very unique type of air composition in this region with contributions of variety of different processes. Due to the nature of the subject, it is probably difficult to give a more clear picture. As one of the most important findings, the unexpected high ozone levels are reported. Of course, it is difficult to explain this effect only from experimental data. However, using model (EMAC), it would be desirable to see how in the region around 150 hPa, the forming of the enhanced ozone levels during summer and September can be understood, or, at least is resolved by EMAC."*

We agree with the reviewer that additional analyses are needed to understand the observed specific trace gas signatures in the broader context of processes generally relevant for the ASMA. Various such analyses were already part of an earlier version of the paper, which however became too voluminous. Thus we split the original draft. Now the current paper focuses on the HALO ESMVal ASMA measurements and their straight interpretation. Placing the measurements into the context of temporal and spatial ASMA variability is now the focus of an accompanying follow-up study (Gottschaldt et al., 2017). The model-based analyses there provide other insights into the underlying processes than the direct interpretation of the measurements. However, a detailed analysis of at least some representative measurements is still needed to understand the results of the more climatological analyses in the follow-up study. Furthermore, the current paper shows that our EMAC simulation reproduces the HALO ESMVal measurements reasonably well. Therefore the simulation may be used for the interpretation of the measurements (this paper), and the measurements provide some ground truth for the simulation-based analyses (accompanying paper). We found it impossible to reconcile those complementary aspects in just one concise paper of reasonable length.

*"Furthermore, the paper tries to report everything one can say without trying to concentrate on the most important points. Thus, I would recommend to get rid of some unnecessary ballast (for some ideas see below). Thus because of these two reason (missing model-related explanation of enhanced ozone levels and too many details) the paper needs a major revision."*

Following the reviewer's recommendations to shorten the text (his/her points 3. to 6.), we removed most of the discussion of POI2/4/5/6 from the main text. POI1/7 are not relevant for the ASMA and have also been moved to an appendix. Now the main text is more concise, but the entire flight is still documented for follow-up studies. The revised main text focuses on POI3, while the other flight segments are only discussed as far as needed to put POI3 into the context of the entire flight. Considering the recommendation of reviewer 1, we also shortened section 6.1

Minor points:

*1. P 3/L 25: "On the contrary" - what do you mean. Contrary to "no decrease" is decrease. Please reformulate*

The revised formulation is: "The in-situ measurements considered in our study also show enhanced CO mixing ratios in the ASMA, but instead of decreased $O_3$ we found significantly increased $O_3$ mixing ratios - relative to the UT air encountered south of the anticyclone."

*2. P 6/L 27: You are talking about streamlines but your never show them. You are only showing trajectories which, in general, do not follow streamlines. (only for a stationary flow streamlines and trajectories are the same lines). After presenting your data (Fig 1) and the vertical and horizontal cross sections through the model (Fig 2) it would be nice to see also the meteorology at e.g. p=150 hPa showing streamlines of the geopotential for few days before the flight.*

We did pay attention to the difference between streamlines and trajectories when writing the paper, but indeed only had streamlines' plots for monthly mean fields in the Supplementary material of this paper and some hand-drawn streamlines in Fig. 2c. A figure with daily streamlines' plots had become part of the accompanying study (Gottschaldt et al., 2017). We thank the reviewer for noting this and added a new figure (now Fig. 3) to the main text, showing a daily sequence of streamlines and geopotential height from 9 days before to 1 day after the HALO flight at the altitude of the measurements.

*3. P 6/L 33: POI2 - here is potential to shorten the text. This flight segment has nothing to do with ASMA*

Most of the discussion of POI2 has been moved to Appendix A.

*4. P 7/L 15: POI4 - too much information. However, the difference between the slow rotation (lower level) and fast rotation (higher level) is an interesting feature.*

Most of the discussion of POI4 has been moved to Appendix A.

*5. P 7/P 8: POI5/POI6 - in my opinion there is no reason to discuss these two flight segments separately. Also one trajectory figure would be enough. The difference between the Iranian and Tibetan mode cannot be seen from your investigation. I would recommend to remove this part of the text*

Most of the discussion of POI5/6 has been moved to Appendix A.
There is an ongoing discussion about different modes or phases of the ASMA (Nützel et al., 2016; Pan et al., 2016). The study of Pan et al. (2016) that came out after our discussion paper distinguishes four phases: Tibetan plateau phase, Iranian plateau phase, longitudinally elongated phase, double center phase. We are aware that a single transect of measurements in the ASMA can contribute to this debate only marginally, at best. Nevertheless we need to discuss the ASMA splitting event that occurred during HALO ESMVal and could be inferred from Figs. 2 + 7, 5e, 5i, S6 in the original draft. It is even more clearly seen in the new Fig. 3 (revised manuscript) and might correspond to the transition from a longitudinally elongated phase to a double center phase in the new nomenclature of Pan et al. (2016). We just use the terms Iranian or Tibetan part/eddy/anticyclone to describe the splitting of one big into two smaller anticyclones, but avoid the words "mode" or "phase" in this context completely.

*6. caption of Fig 3: Please explain only a, b, c, and d panels in this caption. For POI5/6 would be enough to write: same like for POI3*

Done.

*7. P9-11: I agree that the model performs good to represent the in-situ measurements. In the following chapter the tracer-tracer correlations are discussed. It would be nice to see (or only to know) how such correlations are represented in the model. Typically, models do not correctly represent such correlations. Maybe EMAC is better?*

A direct comparison for short flight segments isn't meaningful because of the coarse resolution of the simulation. For instance POI3 is represented by just two points in a simulation with 12 min time stepping. Thus we sampled the entire region throughout September to get an idea on how those tracer-tracer correlations come out in the simulation. However, this rather climatological analysis is not needed for the straight interpretation of the measurements. Thus the tracer-tracer plots from the EMAC simulation are shown in Gottschaldt et al. (2017), together with the ranges covered by the in-situ data.

*8. P11/L3: "...which is carried forward to the related large scale trace gas distributions" – this sentence is not clear for me. Please reformulate*

The revised version reads: "This means in return that a large scale feature like the ASMA is likely to be represented well by the specified dynamics simulation setup, which is also well suited to reproduce the corresponding trace gas distributions."

*9. P11 L15: "might not leave too much freedom" - much too speculative*

We removed that sentence.

*10. P13 L20: Figure 6 is difficult to understand. In particular the marks "N" and "S" are very confusing. I do not see north or south of the ASMA here.*

The "N", "S", "W" and "E" marks have been replaced by a single "SW" mark and a note in the caption that the cuboid is seen from the SW corner (now Fig. 7).

**References**

Gottschaldt, K., Schlager, H., Baumann, R., Cai, D. S., Eyring, V., Graf, P., Grewe, V., Hoor, P., Jöckel, P., Jurkat, T., Voigt, C., Zahn, A., and Ziereis, H.: Working title: Interplay of dynamics and composition in the Asian summer monsoon anticyclone, Atmos. Chem. Phys. Discuss., in prep., 2017.

Nützel, M., Dameris, M., and Garny, H.: Is there bimodality of the South Asian High?, Atmospheric Chemistry and Physics Discussions, 1-30, 10.5194/acp-2016-362, 2016.

Pan, L. L., Honomichl, S. B., Kinnison, D., Abalos, M., Randel, W. J., Bergman, J. W., and Bian, J.: Transport of chemical tracers from the boundary layer to stratosphere associated with the dynamics of the Asian summer monsoon, Journal of Geophysical Research: Atmospheres, 121, 10.1002/2016JD025616, 2016.

| Original manuscript | Revised version |
|---|---|
| Fig. 1 | Fig. 1 |
| Fig. 2 | Fig. 2 |
|  | Fig. 3 |
| Fig. 3 abcd | Fig. 4 abcd |
| Fig. 4 | Fig. 5 |
| Fig. 5 | Fig. 6 |
| Fig. 6 | Fig. 7 |
| Fig. 7 | Fig. 8 |
| Fig. S4 abcd | Fig. A1 |
| Fig. S6 | Fig. A2 |
| Fig. S4 efgh | Fig. A3 |
| Fig. 3 efgh | Fig. A4 abcd |
| Fig. 3 ijkm | Fig. A5 abcd |
| Fig. S4 ijkm | Fig. A6 abcd |
|  | Fig. S1 |
| Fig. S1 | Fig. S2 |
| Fig. S2 | Fig. S3 |
| Fig. S3 | Fig. S4 |
| Fig. S5 | Fig. S5 |
| Fig. S7 | Fig. S6 |
| Fig. S8 | Fig. S7 |
| Fig. S9 | Fig. S8 |

Table 1: Mapping of figures between original and revised manuscript